# The weather behind words. New methodologies for integrated hydrometeorological reconstruction through documentary sources.

Salvador Gil-Guirado [1, 2], Juan José Gómez-Navarro [3], and Juan Pedro Montávez [3]

[1] Interuniversity Institute of Geography, University of Alicante, P.O. Box 99, 03080 Alicante, Spain.
[2] Department of Geography, University of Murcia, Campus de la Merced, 30001 Murcia, Spain
[3] Department of Physics, University of Murcia, Campus de Espinardo (Faculty of Chemistry), 30100 Murcia, Spain.

*Correspondence to*: Salvador Gil-Guirado (salvagil.guirado@ua.es)

**Abstract.** Historical Climatology has remarkable potentialities to produce climatic reconstructions with high temporal resolution. However, some methodological limitations hinder the spatial development of this discipline. This study presents a new approach to Historical Climatology that overcomes some of the limitations of classical approaches, such as the Rogations Method or Content Analysis: the Cost Opportunity for Small Towns (COST). It analyses historic documents and takes advantage of all sorts of meteorological information available in written documents, and not only the severest events, to therefore overcome the most prominent bottlenecks of former approaches. COST relies on the fact that using paper is very costly, so its use to describe meteorological conditions is hypothesised as being proportional to the impact they had on society. To prove the validity of this approach to reconstruct climate conditions, this article exemplarily uses the Municipal Chapter Acts of a small town in south Spain (Caravaca de la Cruz), which spans the 1600-1900 period, and allows reconstructions to be obtained on a monthly basis. Using the same documentary source, the three approaches were used to derive respective climate reconstructions, which were then compared to assess climate signal consistency and to identify possible caveats in the methods. The three approaches led to a generally coherent series of secular variability in the hydrological conditions, which well agrees with previous study results. The COST approach is arguably more objective and less affected by changes in societal behaviour, which allows it to perform comparative studies in regions with different languages and traditions.

**Keywords.** Little Ice Age; Palaeoclimatology; Historical Climatology; Europe; Iberian Peninsula; Documentary Sources; COST; Content Analysis; Rogations.

## 1 Introduction

Understanding climate variability is fundamental to apply a long-term climatic perspective to ongoing global change (IPCC, 2013). Meteorological observations play a prominent role here as they record this variability and enable it to be studied. Despite few early instrumental records starting in the 18th century in industrialised areas (Cornes et al., 2012; Domínguez-Castro et al., 2013; Prohom et al., 2016), the systematic recording of climatic data, globally coordinated by the World Meteorological Organization (WMO), is much more recent. This short period of time precludes the establishment of a robust

statistical characterisation of climate variability and, thus, limits the understanding of those mechanisms behind long-term climate variability. Therefore, it is necessary to use proxy data or direct climate descriptions that allow climate series to be reconstructed beyond short instrumental records (Pfister et al., 2008). Such reconstructions shed light on the relations between social processes and long-term climate variability (Hsiang and Burke, 2013). Climate models and climate reconstructions are the two main tools presently available to climate researchers that enable climate variability to be studied. Both approaches are necessary and complementary as their joint analysis has been shown to constrain uncertainties and shortcomings that would not be otherwise possible (Gómez-Navarro et al., 2015). Climate is a complex phenomenon with wide regional variability that demands a coordinated study (Izdebski et al., 2016) to tackle as many world regions as possible (Giorgi et al., 2009). However, the availability of proxy data to be used in climate reconstructions is very heterogeneous in space and time terms (Consortium, 2013; Consortium, 2017). Each proxy data type has a specific potential and limitations (See Pfister et al., 2008, and IPCC, 2001 Chapter 2, Section 2.3.). In some parts of the world, the specific limitations of each proxy feed one another given its socio-environmental characteristics (Huang et al., 2000), thus large areas worldwide are not represented in palaeoclimatic reconstructions (Luening, 2017). Arid regions are, for instance, generally underrepresented due to limited natural proxies being available (Machado et al., 2011). By way of example, to carry out climate reconstructions by speleothem, specific geological conditions must come into play (Bar-Matthews et al., 1996). Regarding climate reconstruction through ancient tree rings, the existence of appropriate plant species is required (Galván et al., 2014). Finally, the correlation between the presence of ice masses and the feasibility of making reconstructions through ice cores is clear (Thompson, 2000), but most of these requirements are not met in arid and semiarid areas.

In this context, Historical Climatology (hereinafter referred to as HC) is a suitable alternative tool that may bridge this gap because it allows climate reconstruction in locations where natural proxies are scarce (Brázdil et al., 2005; Prieto and García-Herrera, 2009). HC is a very powerful source of insight as the historic documents that record weather-related phenomena prior to instrumental weather data collection are available in many parts of the world (Gil-Guirado et al., 2016; Brázdil et al., 2019). Documentary data are a high-resolution proxy for climate reconstructions (Pfister et al., 2008). Therefore, HC is to be uniquely placed to generate extended datasets that are to be used for climate model validations studies and to provide empirical evidence to further our understanding about the changing nature of climate-society relations over time (Nash and Adamson 2014). A prominent advantage of this research field over other palaeoclimatology approaches lies in its economic-technical requirements being minimal. However, it requires continuous and homogeneous written sources (Brázdil et al., 2005). In Europe, this condition limits the reconstructed period to the last 8 to 10 centuries, depending on the country (Lamb, 1965; Gagen et al., 2006; Brázdil et al., 2010). However, some documentation available in Southeast Asia has allowed researchers to perform reconstructions that date back to more than 20 centuries (Ge et al., 2003; Wei et al., 2014).

All over Europe, where literature on temperatures and precipitation reconstructed through historic documents is abundant, HC is an important source of insight. Studies in Switzerland, Germany, the Czech Republic, France, Hungary, the Netherlands, the British Isles, the Balkans, Portugal, Norway, Italy (for a more detailed analysis of the HC literature in Europe, see Camenisch, 2015, Brázdil et al., 2005 and 2010, and Camuffo et al., 2010) and Spain (Martín-Vide and

Barriendos, 1995; Rodrigo et al., 1999; García-Herrera et al., 2003a; Vicente-Serrano and Cuadrat, 2007; Domínguez-Castro et al., 2008; Fernández-Fernández et al., 2015; Cuadrat, et al., 2016; Tejedor et al., 2018) are examples of well-studied regions. Beyond Europe, other regions where HC has remarkably developed are South America (Prieto, 1985; Prieto and Jorba, 1991; Prieto et al., 2000) and SE Asia (Ge et al., 2003, 2005). However, most regional studies in HC date from the 1980s and 1990s. This period underwent tremendous growth as far as series reconstructed through historic documents emerging are concerned. HC studies have not grown as much as studies have done through other types of proxies. This situation can be explained by two alternative causes: either there are already sufficient reconstructed locations to allow optimum knowledge of climate variability or HC is currently limited by methodological limitations that prevent the catalogue of locations from being extended. The current proliferation of studies that present new reconstructed series from natural sources (Romero-Viana et al., 2011; Nieto-Moreno et al., 2013; Barreiro-Lostres et al., 2014; Tejedor et al., 2016) suggests that methodological limitations are indeed the current bottleneck of HC. Thus new methodologies have to be developed that circumvent current HC limitations to keep it competitive with other fields. Such advances would also greatly benefit Palaeoclimatology given the potential of HC to produce the above-described high-resolution series. Remember that annual, and even seasonal, resolved information is essential to understand how climate variability affects societies (Hegerl et al., 2011).

Spain has one of the largest and most varied documentary heritages in the world which has, unfortunately, not yet been fully exploited for climate studies. For instance, although it has a surface covering more than half a million square kilometres, only 15 reconstructed series currently exist, which simultaneously span more than three centuries and overlap instrumental records. Catalonia (in Northeast Spain) presents good spatial coverage (Martín-Vide and Barriendos, 1995), while the Balearic Islands, and inland and Northwest Spain present notable spatial gaps (See Figure 1, Panel a). Apart from these long series, some initiatives have accomplished reconstructions that span much shorter periods of time, but they do not overlap instrumental data, which precludes the calibration and validation of series against observations (Bullón, 2008; Fernández-Fernández et al., 2015; Alberola Romá, Bueno Vergara and García Torres, 2016 ; Alberola Romá, 2016; Cortizo, 2016). Most of the studies that overlap instrumental data in Spain focus on capital cities (73% of the series), especially in the episcopal see (93% of the series). This is due to the large amount of ecclesiastical and municipal sources used to produce these series, which has a side effect: small populations with no strong ecclesiastical representation are excluded as sources of insight. This scenario contributes to the above-described bottleneck of HC.

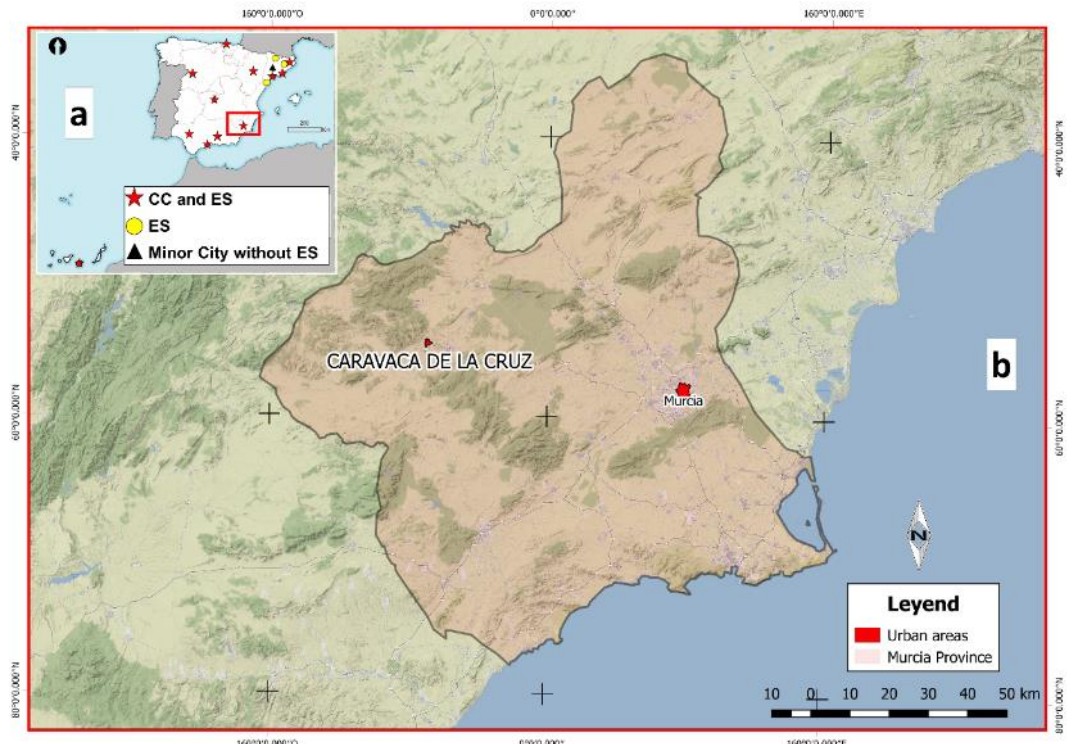

**Figure 1: Location maps of reconstructed series with an instrumental overlap in Spain (Panel a) and the Study Area (Caravaca de la Cruz) (Panel a). Panel a** shows the reconstructed series in Spain that overlap instrumental records. Red stars identify the reconstructed series in Capital Cities (CC) that are also episcopal sees (ES). The yellow circle identifies the reconstructed series in Cities that are Episcopal Sees (ES). Finally, the black triangle identifies the small cities with no capital or Episcopal See. **Panel b** represents the location of the Study Area (Caravaca de la Cruz). *Sources:* Martín-Vide and Barriendos (1995); Rodrigo et al. (1999); García-Herrera et al. (2003a); Vicente-Serrano and Cuadrat (2007); Rodrigo and Barriendos (2008); Gil-Guirado (2013).

Including the sources that current HC methods do not consider entails making major quantitative and qualitative efforts to develop new reconstruction methodologies capable of integrating more world locations (Pfister, 2014) and making the obtained series comparable in time and space terms (Nash & Adamson, 2014). This is, however, a difficult task and one which poses numerous questions that need addressing, such as: What is the most appropriate methodology for each location? Does it depend on the targeted period? Are there complementary and alternative methods? Answering these questions is no easy task, and it is necessary to analyse the sensitivity of different methods as this knowledge will allow the methods for each application to be optimised. By making a prominent effort in this direction, Neukom et al. (2009) used the pseudo documentary source concept to confirm how good complementarity exists between the series reconstructed with different kinds of historical sources, e.g. newspapers and official documents. However, the aim of such an analysis was a methodological validation for the same place using the same documentary sources, but by different methodologies, which is infrequent in the HC literature

There are currently two main HC methods that have been used to produce continuous series with historic documents from the old Spanish Empire. On the one hand, the rogations method (Martín-Vide, & Barriendos, 1995; Barriendos, 1997;

Rodrigo and Barriendos, 2008; Domínguez-Castro et al. 2008) maps ceremonies asking God for rain (pro-pluvia) or to stop rain (pro-serenitate) in precipitation indices; on the other hand, the Content Analysis (Prieto et al. 2003; Prieto et al. 2005), which maps the precise wording used in historic documents to describe meteorological events in numerical values (typically rain, temperature and wind). The application of these two methods is subject to one important factor, i.e. the amount of available documentation. In this sense, Spain is a good target for such studies because important sources of historic documentation are currently available and are more abundant than in, for instance, South American where historic vicissitudes have caused a larger part of this documentation to be lost (Gil-Guirado, 2013). However, when the amount of documentary sources is very large, consultations of the full material become too complex and time-consuming and, therefore, the selection of the most suitable sources is necessary (Brazdil et al, 2005). In this circumstance, the most appropriate method is the rogations (RO) method as it enables highly robust reconstructions using a fraction of the total documentation (Barriendos, 1997; Gil-Guirado, 2013). This condition explains why the RO method is the most widely used one in the HC literature in Spain. Conversely, when the amount of documentary sources is scarce, it is necessary to consult the documentary sources in full detail. This forces researchers to use documents from heterogeneous sources to gain an advantage from every piece of climate information that they might contain. In this circumstance, the most appropriate method is the content analysis (CA) as it allows documentary sources of various kinds (civil, religious, private, etc.) to be analysed by a common methodology to enable robust series to be obtained (Prieto et al. 2003). This situation has resulted in the CA method being the most widely used one in South American countries (Prieto, Herrera & Dussel, 2000; García-Herrera et al. 2008; Neukom et al. 2009; Prieto & García-Herrera, 2009).

Bearing in mind this review on the current status of HC, the objectives study are to:

1. Describe and validate a new suitable methodology to reconstruct climatic series in small towns through historic documents (the so-called Cost Opportunity for Small Towns method-COST)
2. Simultaneously apply the RO, CA and COST methods to the same historic document by analysing the sensitivity of the results to the chosen method by thus characterising the uncertainties and robustness of this new approach
3. Determine if complementarity between methods exists for it to be used to bridge gaps when the requirements for applying one of these three methodologies are not met.

As a testbed to apply the methods, we took the data from a small size population in Southeast of Spain (Caravaca de la Cruz, 25,633 inhabitants in 2017) (See Figure 1, Panel b) over the 1600-1900 period. The semiarid conditions (around 380 mm annual average) and mild temperatures (about 16°C annual average) render the precipitation and water availability as the determining factors for human activity (DeMenocal, 2001: 667). Therefore, precipitation is the target variable for reconstruction.

This paper is arranged as follows: Section 2 presents and discusses historic sources. Section 3 focuses on the thoughtful description of the three methodologies. Section 4 presents the reconstructions carried out and compares the results. Finally, Section 5 concludes the paper by making the main remarks.

## 2 Documentary sources

The previous analysis of all documentary sources is a necessary step in every HC study (Brazdil et al, 2005). The next step is to objectify the obtained information; i.e. convert historic data into climate data (Glaser, 1996: 57).

For this paper, we consulted the Actas Capitulares (Municipal Chapter Acts) of the Concejo (City Council) of Caravaca de la Cruz (henceforth referred to as Caravaca). In all, we consulted 50,388 sheets of paper for the 1600-1900 period. Notwithstanding, it was not possible to bridge some documentary gaps (from 1820 to 1823, and 1891 to 1892).

The Municipal Actas Capitulares (MCA) of City Councils (CCO) are the most useful source for climate studies in Ibero-American countries (Metcalfe et al. 2002). After the foundation of each new town, the CCO was established for its government. Councillors and Mayors (Capitulants) held weekly meetings during which all town management matters were analysed[1]. It was mandatory to leave a written record of meetings and records in the municipal archive for possible future claims or consultations. A public scribe was responsible for transcribing all the information in a MCA book. In these books, the date, the participants, the topics covered, the specific contributions of each participant and any reached agreements were recorded (see Figure 2 a, for an example of this type of documents). This arrangement remained intact for centuries (see Figure 2 b, for different MCAs throughout the study period). MCAs were official documents that were countersigned on official State paper. By law, the central State obliged this arrangement to be maintained. Therefore, these MCAs were extremely similar in terms of their structure and composition. However, the rule about holding weekly meetings was not strictly followed through, and the number of meetings was more related to urgent themes than to administrative matters. In other words, the dynamics of CCO meetings responded to towns' day-to-day problems. For this reason, some weeks went by when no meetings were held, while several meetings could have taken place in just one week. In any case, rules about having to hold weekly meetings explain that there was no seasonality in the number of MCAs used each month, which has been evidenced in other areas of Spain (Pérez, 1987; Gutiérrez, 2005), but the most important point is that there was no time of the year when meetings had to be put off. Hence the importance of matters and the situation of a given year explain the bigger or smaller quantity of paper used in a year, but also ensured that whenever any extraordinary event occurred, like scarce rainfall or heavy rain events, and not matter what the date, the CCO would meet to deal with any particulars.

This makes the MCA book a kind of Official Gazette in which all issues that affected towns were recorded (Barriendos, 1999). In the MCA, all environmental aspects that influenced the population were comprehensively treated. In this way, all anomalous events were collected in great detail. Therefore the existence of a MCA with no references to meteorological information implies that the climate and environmental situation were regarded as being normal because when the climate thresholds accepted by a society are not exceeded, there is no news (Prieto, Herrera and Dussel, 1999). Taking into account these described characteristics, MCA books are the primary council source to offer direct weather data, water-dependent data and phenological data (Bradley & Jones, 1992: 12).

---

[1] These were the ordinary meetings. If some extraordinary event, such as floods, plagues and epidemics, took place they held an extraordinary meeting. Therefore depending on the needs, meetings could be held on any day of the year.

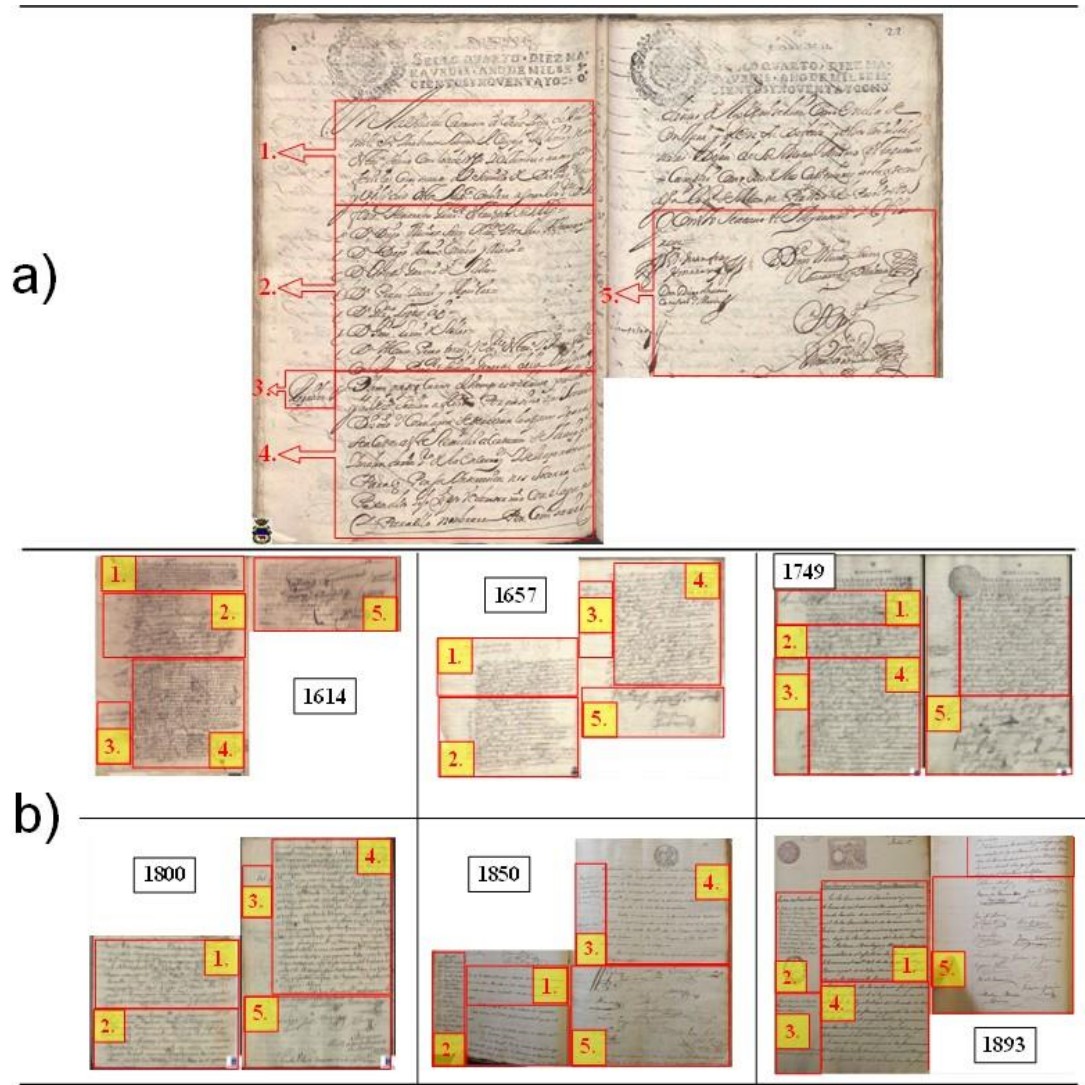

**Figure 2: The Caravaca CCO meeting sample (April 18, 1698) (Panel a) and different examples of MCA throughout the study period (Panel b).**

The main parts of a MCA are marked : 1-Date, place and type of meeting (Ordinary or Extraordinary) ; 2-Full name of those attending the meeting; 3-Short summary of the topics discussed; 4- Topic memorandum and attendees' contributions; 5- Attendees' signatures. *The text in Panel a,describes a ceremony to ask God for rain (pro-pluvia): "*Dijeron que por cuanto el tiempo es adelante y no llueve y los vecinos se hallan afligidos, porque si no nos socorre Dios mío con el agua se perderán los panes y para que se acuda a este remedio acordaron se traiga a la imagen de Nuestra Señora de la Encarnación y se le haga novena, para que por su intercesion nos socorra su bendito hijo con el agua y para ello nombraron por comisarios a los Señores Don Francisco de Quesada y Don Alfonso de Sajosa. Y en este estado dijeron que por cuanto Don Alfonso de Sajosa se halla en el campo y no puede venir y por no estar delante el Señor Don Francisco de Quesada parece se podrá excusar y para obviar cualquier excusa acordaron se repartan suertes entre los capitúlales que están y toco a los Señores Don Francisco de Quesada y Ginés de Gadea*". Source:* the Carmesi Project.

*This text is written in Old Spanish. An approximate translation is the following: "They said that the season is very advanced and it has not rained and the population is afflicted, and unless God helps us with rain the crops will be lost. Therefore, they agreed to bring the image of "Our Lady of Encarnación" and to celebrate nine masses praying so that She asks to her Son to help us with rain. For this reason, Don Francisco de Quesada and Don Alfonso de Sajosa were appointed as commissioners. And they said that it was necessary to make a raffle among the capitulants because Don Alfonso de Sajosa was working in the fields and Don Francisco de Quesada was not in this meeting. After making this raffle, Don Francisco de Quesada and Ginés de Gadea were elected as commissioners."*

## 3 Methodology

Through the different applied methodologies, continuous data series were obtained from 1600 to 1900 on a daily basis. For the data analysis, daily values were aggregated on monthly, seasonal (Winter: December, January and February; Spring: March, April and may; Summer: June, July and August; and Autumn: September, October and November) and annual scales. The analysis of drought and extreme rainfall series was carried out separately to create seasonal and annual series for droughts and extreme rainfall, respectively.

### 3.1 The rogations method

Rogations are liturgical acts in which the Catholic Church asks God for rain (pro-pluvia rogations, hereinafter referred to as PPR) or to stop rain (pro-serenitate rogations, hereinafter referred to as PSR). Catholic countries present a high incidence of such ceremonies (Espín-Sanchez and Gil-Guirado, 2016). In Latin American countries, this historical context and the absence of more precise proxies explain the proliferation of academic articles using RO data to reconstruct weather patterns (Martin-Vide and Barriendos, 1995; Alcoforado et al., 2000; Garza Merodio, 2002; Domínguez-Castro et al., 2018). Spain is a country where remarkable results have been obtained following this method (Martín-Vide & Barriendos, 1995; Rodrigo and Barriendos, 2008; Domínguez- Castro et al., 2008).

Catholic ROs are religious rites performed for specific purposes, such as earthquakes, droughts, heavy rain and floods. PPR are the most extended and frequent form of them. Since the 9[th] century, the RO procedure has been strictly regulated by the Vatican (for more details, see Espín-Sanchez and Gil-Guirado, 2016).

In this way, implementing PPR has been consistent over time in all Catholic territories. The PPR process usually comprises the steps below (Garza and Barriendos, 1998):

i.   The local government (civil authorities) receives a RO request from farmers and decides to ask the religious authorities for a RO request
ii.  The religious authorities receive the RO request from the civil authorities, and then the religious authorities decide the RO date and ceremony type
iii. The ceremony is performed
iv.  When rain occurs, the religious authorities decide to give a thanksgiving mass.

The institutionalisation of the RO process allows us to differentiate between different RO levels, depending on the performed ceremony type. The RO cost has increased in line with an augmented RO level (Espín-Sanchez and Gil-Guirado, 2016). We adapted the methodology proposed by Martín-Vide and Barriendos (1995) by classifying PPR into five different levels of drought intensity, from level 1 (the weakest drought) to level 5 (the severest drought). The different levels are as follows:

1. RO masses held in church
2. RO masses held with figures of saints or virgins exhibited in church
3. Popular processions through city streets with figures of saints or virgins
4. Popular pilgrimages to a sanctuary outside the city, carrying figures of saints or virgins
5. Water body immersion (river, well or fountain) of the figures of saints or virgins

In addition to PPR, PSR are also recorded, but much less frequently, as reported below. Accordingly, PSR are also classified into five increasing levels of rain intensity:

1. Thanksgiving masses: masses held to thank God for rain arriving
2. RO masses held in church or spells against storms
3. Popular processions with figures of saints or virgins exhibited in church
4. Popular saints or virgins exhibited from the church tower
5. Popular processions with figures of saints or virgins exhibited and other uncommon ceremonies in PSR rogations

In the ceremonies of level 3 or higher, with two images (statues of saints or virgins) or more, we add another level point for each additional image (e.g. A PPR-Procession with three statues of saints scores 5 points, with 3 points for the profession and 2 points for additional saints). Such strengthened ceremonies were held when the adverse situation persisted, despite them having prayed for a long time (See Figure 3).

The way to code a PPR or a PSR follows some steps (see the example in Figure 3). Once a RO is detected in the CA, we read it to identify if it was an RO for shortage of rain (PPR) or an RO for rain to stop (PSR). The text always explicitly specifies this information because it provides an explanation about the reasons why an RO was organised ("does not rain" or "shortage of rain" are the most frequently found wordings when requesting a PPR). Then we analyse the liturgy used in the RO to determine its level. By following the example in Figure 3, an agreement is reached at the CCO to carry a figure of the Virgin Encarnación to perform the RO in church, so its intensity level is Level 2. With this information, we will obtain the date the RO was held, the reason for or type of RO (Drought-PPR) and its level (Level 2). So with this information classified by date, type and level, it is easy to aggregate data later to offer data on a monthly, seasonal or yearly basis.

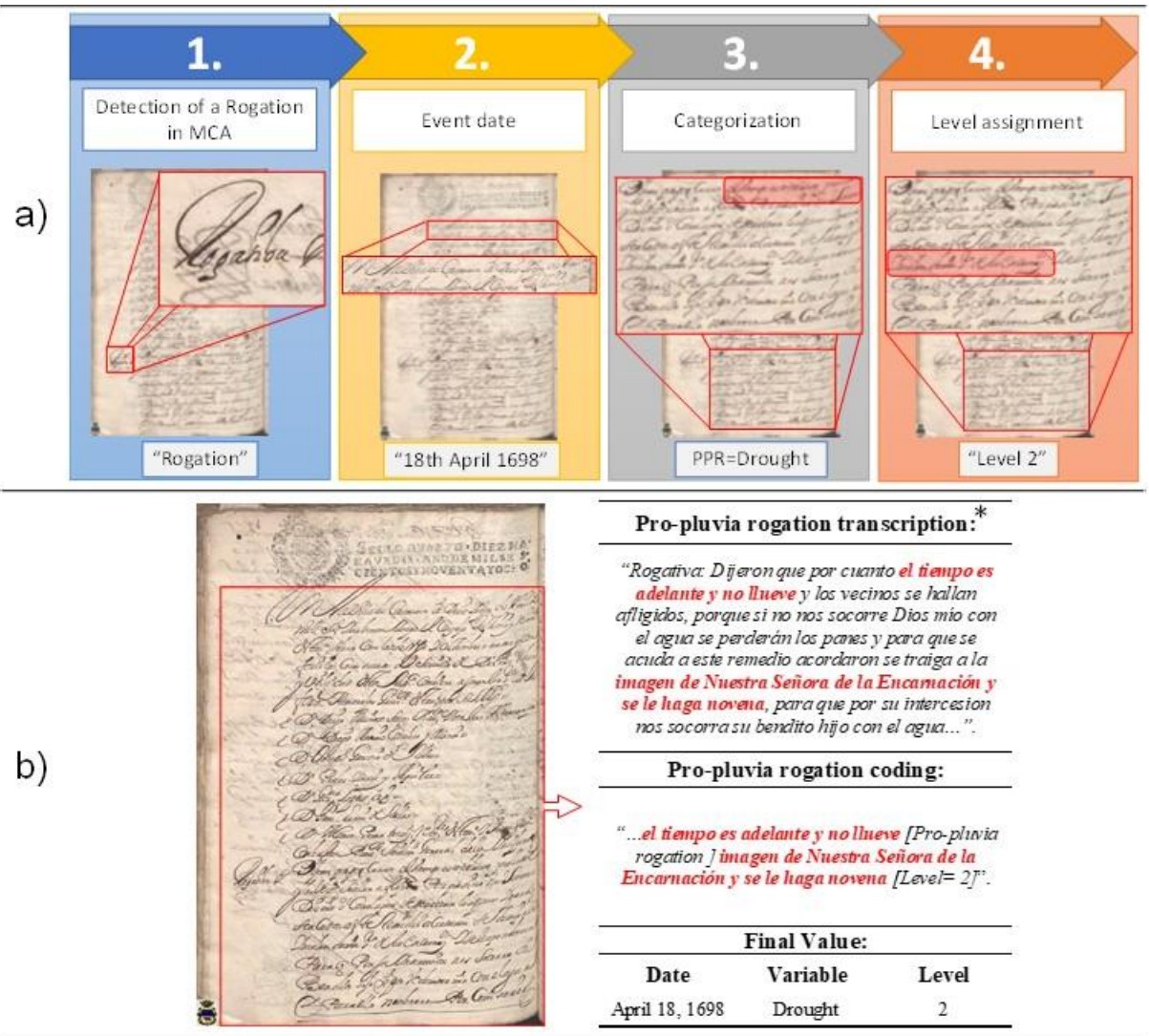

**Figure 3: RO method by step (Panel a) and encoding example of the RO method (Panel b).** This particular example refers to a Pro-Pluvia RO on 18th April 1698, so the reconstructed variable is drought. Source: the Carmesi Project.

*\*This text is written in Old Spanish. An approximate translation is the following: "They said that **the season is very advanced and it has*
5 *not rained** and the population is afflicted, and unless God helps us with rain the crops will be lost. Therefore, they agreed to bring **the*
*image of "Our Lady of Encarnación" and to celebrate nine masses** praying so that She asks to her Son to help us with rain."*

### 3.2 Content Analysis

One of the main problems which Palaeoclimatologists encounter when transforming the qualitative information contained in documental information into quantitative data is homogenisation when describing in writing a climate-type phenomenon.

The official nature of documents is an advantage for such cases because the patterns and ways to act over time are repeated, and some terms that refer to possible climate events are constantly repeated. A CA is a research technique used to identify the meaning given to full written texts by taking into account the historic, and socio-cultural contexts in which they were

drafted (Bardin 1986). The intention of a CA is to quantify a series of parameters to carry out statistical analyses. This is a matter of analysing the frequency with which terms were used and the intensity of certain key concepts, defined as time units or in the space they occupy in the document. The use of CA in climate studies is relatively recent and not yet widespread. Moodie and Catchpole (1975), and later Baron (1982), pioneered the CAs application in HC. Recently, the works of Prieto

(Prieto et al. 2003; Prieto et al. 2005) have promoted this technique by showing its validity for environmental history. Some studies have used this technique in Iberia (Domínguez-Castro, García-Herrera and Vaquero, 2015), but only for a limited number of years and without it overlapping instrumental data.

CA studies must follow several steps to be conducted. Bardin (1986: 71) distinguishes the following: 1) the pre-analysis, when the documents to study are selected, the hypothesis is formulated and objectives are set; 2) the use of material, when

documentation is encoded; 3) data processing, inference and interpretation.

We used the Caravaca MCA according to the present work objectives and hypothesised that the different linguistic expressions employed by contemporary people to describe a climate event, and to report the intensity, duration and direction of this event.

Documentation encoding was done on the topics in the MCA, which are susceptible to contain information about droughts

and extreme rainfall. They are mainly the following:

      − PPR and PSR topics

      − Information about crops (mainly cereals and wine)

      − Information about food prices (bread, wine, meat, etc.)

      − Cattle status topics

− Measures taken against floods and droughts

      − Topics about not paying taxes and the reasons

      − Information on the state of roads and communications

After locating all the expressions containing information about droughts and extreme rainfall, this information is classified according to date (Figure 4, Panel a, Step 1 and 2). Next all the information about the news is read to identify the exact

expression that defines the climate event (Figure 4, Panel a, Step 3). As Prieto et al. (2005: 45) point out, many expressions are synonymous, so the initial number of expressions detected to define a drought (341 different expressions) is summarised in 49 Registration Units (RU) for Droughts. For extreme rainfall events, the 105 different expressions are summarised in 42 RU.  To identify the exact meaning of each RU, Spanish language dictionaries covering the entire study period were reviewed[2]. As seen in similar studies in the Spanish language (Prieto and Jorba, 1991), no evident changes exist in the use of

language to describe climate anomalies during the study period.

---

[2] Covarrubias, Sebastián De: Tesoro de la lengua castellana o española. Madrid, Luis Sánchez, 1611. Gaspar y Roig: Biblioteca Ilustrada de Gaspar y Roig. Diccionario enciclopédico de la lengua española, con todas las vozes, frases, refranes y locuciones usadas en España y las Américas Españolas [...] Tomo I. Madrid, Imprenta y Librería de Gaspar y Roig, editores, 1853; Gaspar y Roig: Biblioteca Ilustrada de Gaspar y Roig. Diccionario enciclopédico de la lengua española, con todas las vozes, frases, refranes y locuciones usadas en España y las

The next step (Figure 4, Panel a, Step 4) consists in assigning a numerical value and classifying it according to its intensity (greater or lesser drought intensity, and greater or lesser extreme rainfall intensity) to the various RUs. To assign a value to each RU, we considered all the RUs for each variable (drought and extreme rainfall) by establishing a range between 1 and the total number of RUs of this variable. The RU with the lowest intensity is assigned to a value of 1, whereas the RU with

5 the most intensity is associated with a number that equals the total number of RUs. Between range 1 and the total number of RUs of that variable, the rest of the RUs were classified from less to more intensity. Finally, values were normalised to provide a value between 0 and 1.

To determine the position of each RU within its range of the variable, we categorised firstly the general descriptors of the phenomenon, and secondly adjectives and adverbs. In this way, descriptors determined intensity (Prieto and Jorba, 1991:

50).

For example, we assigned the value of 1 to the RU "alguna falta de agua" ("some water is lacking") because it describes mild drought. We assigned a value of 49 to the RU "extrema necesidad del agua" ("extreme need for water") as it describes the worst recorded drought. Thus by using values between 1 and 49, the remaining 47 RUs were classified from less to more drought intensity. In the example, (see Figure 4, panel b) the expression "the season is very advanced and it has not rained"

found in information about a pro-rain RO, is included in the "No rain" RU, which is located in position 10 of the 49 drought RU, with an intensity value of 0.2.

Américas Españolas [...] Tomo II. Madrid, Imprenta y Librería de Gaspar y Roig, editores, 1855. Real Academia Española. Diccionario de la Lengua Española, vigésima segunda edición. Available in: http://buscon.rae.es/diccionario/drae.htm.

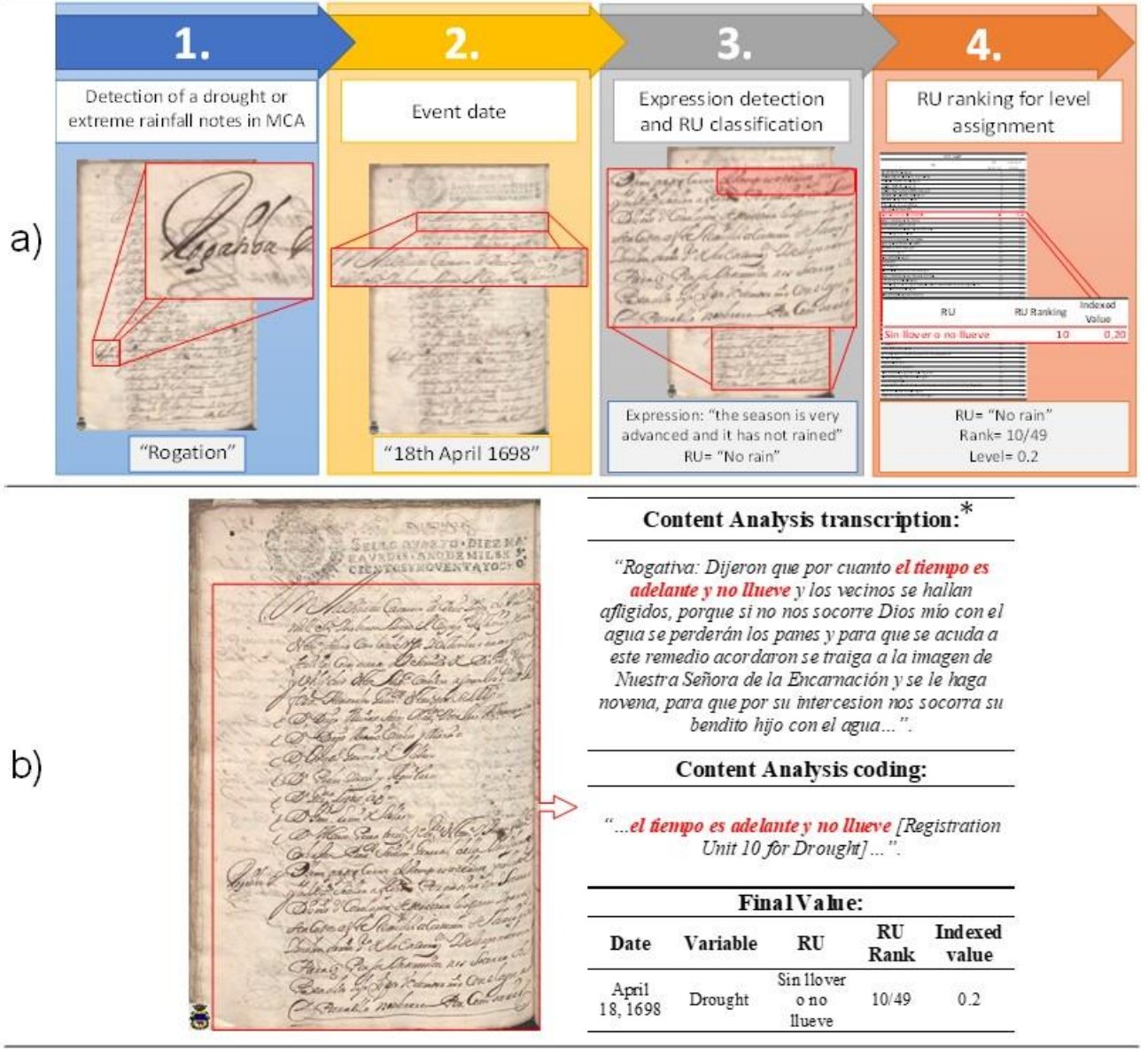

**Figure 4: Content Analysis method by step (Panel a) and an example of the encoding of the Content Analysis (CA) method (Panel b).** The coded source is the same as in Figure 2 to emphasise how the three different approaches are applied in practice. This particular example refers to a PPR on 18 April 1698, so the reconstructed variable is drought. *Source:* the Carmesi Project.

*\*This text is written in Old Spanish. An approximate translation is the following: "They said that **the season is very advanced and it has not rained** and the population is afflicted, and unless God helps us with rain the crops will be lost. Therefore, they agreed to bring the image of "Our Lady of Encarnación" and to celebrate nine masses praying so that She asks to her Son to help us with rain."*

### 3.3 Opportunity Cost for Small Towns (COST)

We propose a new methodology to be applied to small towns, the Cost Opportunity for Small Towns (hereinafter referred to as COST). This method provides comparable results with the CA and RO methods as it is also based on a text analysis.

However, it exploits the volume occupied by the text, rather than subjective linguistic terms. In fact if the intensity of certain key concepts is analysed in a CA, the COST approach analyses the same key concepts, but in units of the space occupied in the document; i.e., we obtain the amount of MCAs that cover extreme rainfall and drought matters.

The underlying hypothesis is that the larger the amount of paper used to describe a climate event, the greater the intensity
and the longer the duration of this event. The requirements of this methodology are as follows:

  i.    Availability of continuous and homogeneous documentary sources
  ii.   The use of paper must have some kind of economic limitation. This condition implies that referring to a particular issue has an opportunity cost for not using that paper for another topic.

The use of paper in MCAs was affected by the taxes imposed by the Central State through the "Papel Sellado" (Sealed
Paper). The "Sealed Paper " in Spain was a special type of paper with a royal seal to improve the reliability of public deeds and to contribute to the cost of the monarchy (Rodriguez, 1996). This paper was an expensive material that incurred significant expenditure for CCOs. From the 17$^{th}$ century in Spain, needs for paper exponentially increased and, at the same time, the Spanish paper industry could not meet paper demands (Hidalgo-Brinquis, 2006). This situation forced CCOs in small towns to take austere measures for using paper. Thus the second above-described condition is met. Moreover, all the
topics discussed during CCO meetings incurred a high cost for towns. Firstly, a series of direct expenses were necessary during each meeting; e.g. the food and drink that the participants partook, wax for lighting, the participants' travelling costs, paying the salaries of public scribes, stewards, etc. Secondly, each topic was preceded by a previous work to analyse the situation and to collect the necessary information (Martínez, 1996).

From these requirements, we quantified the amount of paper (percentage of MCA used during CCO meetings), including any
information about drought and rain. Thus the studied variables are the same as for MCA. The encoding was done on the same MCA topics as in the CA (see the previous section).

The official nature of MCAs determines that neither their structure nor their composition has varied during the study period. However, it is true that the quantity of paper used each year presents wide interannual variety. This variability in the amount of "Sealed Paper" may have something to do with the need to deal with more or fewer matters during a given period. It
might also be related with CCOs having bigger budgets because, if more money was available, CCO could have posed fewer restrictions on the use of paper. Nonetheless, the variability in the amount of paper used each year shows no trend with time (see Figure A1 in Appendix A). Moreover, no statistical correlation was found between the quantity of "Sealed Paper" used each year and the percentage of this paper used to discuss droughts and extreme rain events (see Figure A2 in Appendix A). Moreover in extreme years in terms of the quantity of "Sealed Paper" used (regardless of this amount being excessive or too
small), no extreme percentages of the "Sealed Paper" used to discuss droughts and extreme rain events were found.

As for the seasonality of the employed "Sealed Paper", Royal Ordinances expected council meetings to be held once weekly, irrespectively of the time of year. For this reason, the use of "Sealed Paper" presented no seasonality and the amount of used paper was similar for all the months of the year. Nonetheless, less paper was used in the months of November and December in Caravaca (see Figure A3 in Appendix A).

To expedite the COST implementation, digitising documentary sources is recommended. In this work, the Caravaca MCAs are available on the Carmesi Project[3] website for the 1600-1699 period. For the 1700-1900 period, the authors had access to MCA photographs. These photographs were taken during a collection data campaign in the Caravaca Municipal Archive.

The steps to follow to implement this methodology were as follows (see Figure 5, Panel a): 1. Detect the place where there is information about droughts or extreme rainfall events; 2. Date the event: 3. Count the total amount of "Sealed Paper" used during this meeting to talk about drought or extreme rainfall events; 4. Calculate the percentage of this information of all the "Sealed Paper" used during this meeting and of all the "Sealed Paper" used that year.

The encoding procedure is as follows (see Figure 5, Panel b): we divide each MCA sheet into 200 cells. In this way, each CCO meeting is broken down into a number of cells and a date. Then we quantify the number of cells occupied by the studied variables (if any). At this point, we have the dates, the amount of total cells and the amount of cells occupied by each variable. Finally for each meeting, we aggregate the total cell number and the number of cells in each variable, and then we calculate the percentage occupied by each variable for each meeting. In the example provided in Figure 5 (panel b), we can see that a meeting was held on 18th April 1698, during which almost six sheets of "Sealed Paper" were used (a total of 1,140 cells of papers). Of this amount of paper, less than one sheet (150 cells) was used to talk about a matter related with the problems that resulted from the drought taking place at that time; 150 cells represent 13.16% of all the cells of this meeting. If we bear in mind that 19,600 cells of "Sealed Paper" were used (98 pages of MCA used by the Caravaca CCO in 1698), we obtained 0.77% of all the "Sealed Paper" employed in 1698 to discuss the April drought. In this work, we used values related to annual values; i.e., for every month, we represent the percentage of MCAs employed to discuss drought and extreme rainfall events in relation to all the annual MCAs.

---

[3] The Carmesi Project website: http://www.regmurcia.com/servlet/s.Sl?METHOD=

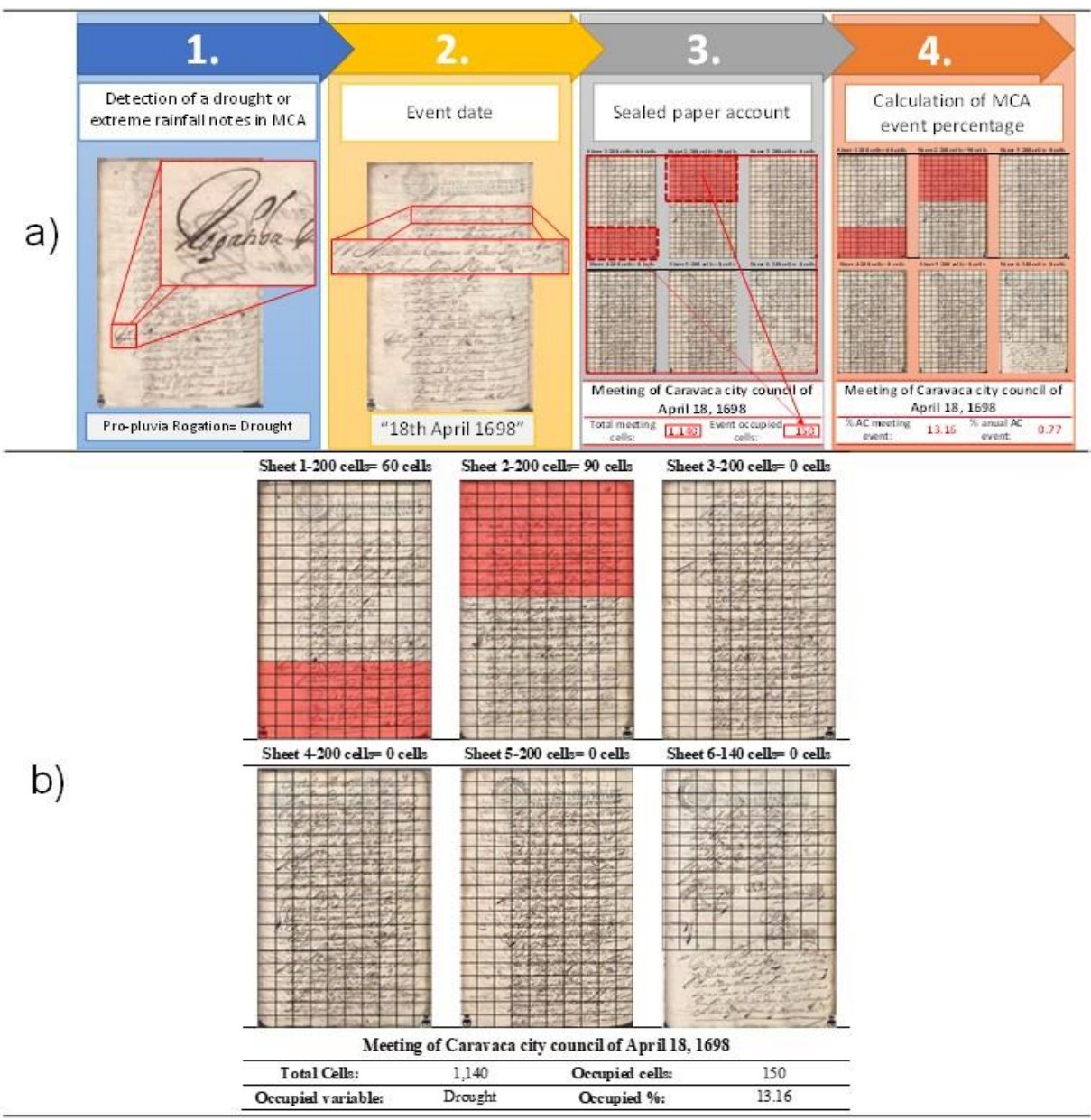

**Figure 5: COST method by step (Panel a) and an example the COST method encoding (Panel b).** The coded source is the same as in Figure 3 and 4 to emphasise how the three different approaches are applied in practice. This particular example refers to a PPR on 18 April 1698, so the reconstructed variable is drought. *Source:* Carmesi Project.

5  **3.4. Climate analysis of the series of drought and extreme rainfall.**

The analysis of the consistency of the data involves homogeneity tests that allows to find abrupt changes, as well as to discern whether such changes have either a climatic or a methodological origin. This analysis is carried out for the drought and extreme rainfall reconstructed series separately. For each variable we have applied standard homogenization tests to

both, the frequencies of appearance, and the intensities of annual values. We use several non-parametric tests to determine possible changes in the data. Although robust, they all have the disadvantage of being able to identify a single jump in the series. The homogeneity tests include the Pettitt Test (Pettitt, 1979), the Standard Normal Homogeneity Test (SNHT) by Alexandersson (1986) and the Buishand rank test (Buishand, 1982). Due to their similarities, the results are very similar

among tests. However, the Pettitt test is the most widely disseminated, while the SNHT has a mathematical predisposition to detect jumps near the beginning and end of the series. On the other hand, the Buishand rank test is more sensitive to non-homogenities in the middle of the series (Renom Molina, 2009). The XLSTAT software (Addinsoft, 2018) has been used for the application of these tests.

We also analysed the low frequency variability of the reconstructed indices by analysing the 10-, 20- and 30-year running

means, and by the Mand-Kendall test with a 95% confidence level. The XLSTAT software (Addinsoft, 2018) was used to run these tests.

## 4 Results

The number of extreme rainfall and drought events reconstructed by the CA and COST approaches is equal, while this number is always smaller in RO. This is due to the fact that whenever documents reflect drought or excess rainfall events,

this information can be classified according to CA and COST. However, the RO method only allows climate information to be collected when these events lead to liturgical processes of ROs. Therefore, all the RO data are included by both CA and COST, and also coincide temporally with one another, while some data obtained by applying the CA and COST data are missed by RO. In our study, 7.1% of the months between 1860 and 1900 take non-zero values in RO, whereas this value reaches 12.9% of the months in CA and COST. This indicates the better sensitivity of RO to hydrometeorological extremes.

However, this asymmetry depends on the variable being reconstructed. Hence major differences appear between drought events (6.4% of the months in RO *versus* 8.5% in CA and COST) and extreme rainfall events (0.6% of the months in RO *versus* 4.4% in CA and COST) (See Table 1).

This has two main consequences: (1) reconstruction methods are more sensitive to drought events than to rainfall. DeMenocal (2001: 667) argues that this lack of rain is the factor that most strongly affects human activities in semiarid

regions; (2) the applicability of RO to reconstruct extreme rainfall events is not as clearly established as it is for drought. This is because the religiosity associated with climate is more tightly bound to climate situations with a stronger potentiality to produce long-term impacts on society, which is the case of droughts in semiarid climate zones (Espín-Sanchez & Gil-Guirado, 2016).

Table 1 summarises the number of months with drought and extreme rainfall events during the 1600-1900 period for all

three methodologies. The average and standard deviation values are also shown for descriptive purposes only. RO has an average PPR of level of 2.4, whereas PSR has a level of 1.5. CA indicates an average intensity of 8.7 RU for drought and an average intensity of 11.3 RU for extreme rainfall. Finally, COST leads to 0.9% and 1% of paper used to discuss each event

type on average, respectively. In this way, we can see how the average extreme rainfall values are above those for lack of rainfall in CA and COST, whereas this situation is reversed for RO. This further illustrates the less sensitivity of RO to detect the previously discussed extreme rainfall. The variability of the indices obtained by the three methods differs and depends on the variable being reconstructed. While the average deviation in droughts indicates similar values across all methods, with extreme rainfall this value is lowest for COST and highest for CA. CA tends to overestimate extreme rainfall events.

| | Drought | | | Extreme rainfall | | |
|---|---|---|---|---|---|---|
| | **RO** | **CA** | **COST** | **RO** | **CA** | **COST** |
| **N** | 228 | 301 | 301 | 22 | 154 | 154 |
| **%** | 6.44 | 8.50 | 8.50 | 0.62 | 4.35 | 4.35 |
| $\overline{X}$ | 2.39 | 8.72 | 0.85 | 1.50 | 11.33 | 0.96 |
| **S** | 1.84 | 7.82 | 0.80 | 0.96 | 10.08 | 1.09 |
| **Average deviation** | 0.74 | 0.71 | 0.71 | 0.71 | 0.80 | 0.61 |

**Table 1: Basic statistics of the reconstructed series by the three methods for both droughts and extreme rainfall, respectively.**
N refers to the number of months where the series differs from zero during the 1600-1900 period, excluding data gaps (1820-1823 and 1891-1892). Percentage (%) indicates the fraction of months including an event. $\overline{X}$ and S represent the mean and standard deviation of the series, respectively. The average deviation is a dispersion metric calculated as the average of the absolute difference between a given month and the mean of the series.

## 4.1 Annual cycle and agroclimatic context

Relating the annual cycle of the reconstructed series with the water variability of the agricultural system allows their robustness to be analysed. Indeed the strong decoupling between reconstructed series and water agricultural requirements could invalidate the use of historic documents to reconstruct climate (Gil-Guirado et al., 2016) because it would indicate that the social water demand is not related to water scarcity.

With droughts, the averaged frequency and intensity present an absolute maximum in spring, together with a secondary one in autumn, and a minimum in summer (seen more clearly in RO) (see Figure 6, a). This annual cycle can be directly related to the distribution of precipitation in Caravaca given its Continental Mediterranean character. Spring is the rainiest season, followed by autumn, but summers are dry. The agricultural specialisation in this region, based on cereal crop growing (mostly wheat and barley)[4] is the result of adaptations to this seasonality. Autumn rainfalls are especially critical for plant cereals, while spring rain is important for optimal growth. Conversely, summer is the harvesting season, so rainfall is not necessary in these months, and can prove inconvenient if excessive. This explains why references to droughts in summer are rare in historic documents. When severe drought occurs in spring, summer could suffer issues that result from water shortages in spring. This might explain why the differences between RO and the other two approaches are especially large in summer. In these months, dry conditions may lead to water management requirements, which are recorded in historic documents, but did not lead to any RO ceremony. In summary, the intensity and frequency of the annual cycle specified in documents well reflect the actual precipitation regime in Caravaca. This indicates that when water demand occurs in the

[4] The Cadastre of the Marques de la Ensenada in 1756 pointed out that the main agricultural products of Caravaca were wheat, barley and rye. Ref. General File of Simancas: AGS-CE-RG-L463-324.

months when water is lacking, it can have the strongest social impact and, therefore, demonstrates the ability of these reconstructed series to reflect actual past climate conditions, particularly drought variability.

For extreme rainfall, major differences appear between the frequencies of RO and the other two methods (see Figure 6, b). RO shows a clear spring maximum, which counteracts the season with the heaviest rains in Caravaca, whereas CA and COST do not present such a clear maximum. However, the few extreme rainfall events in RO (N=22) reduce the reliability of the frequencies reconstructed by this approach. It is noteworthy that for rainfall, the reconstructed series reflects the inherent heterogeneity of precipitation to the SE of the Iberian Peninsula. Therefore in spring and winter, precipitations are associated with frontal systems, with potential damage because water accumulates over several days. In autumn, severe flash floods are associated with convective storms that are frequent and have a strong impact (Martin-Vide, 2004). Although convective precipitations are not exceptional in summer, which determines that CA and COST increase rainfall intensity in June, this is the cereal harvesting month and, therefore, excessive rain can lead to major economical loss, which is reflected in the documents. Both CA and COST indicate high average intensities in the rainiest spring and autumn months. Nevertheless, lack of a clear annual cycle suggests that the reconstructed precipitation spectrum is limited mostly to flash flood episodes, and also to events associated with the presence of frontal systems.

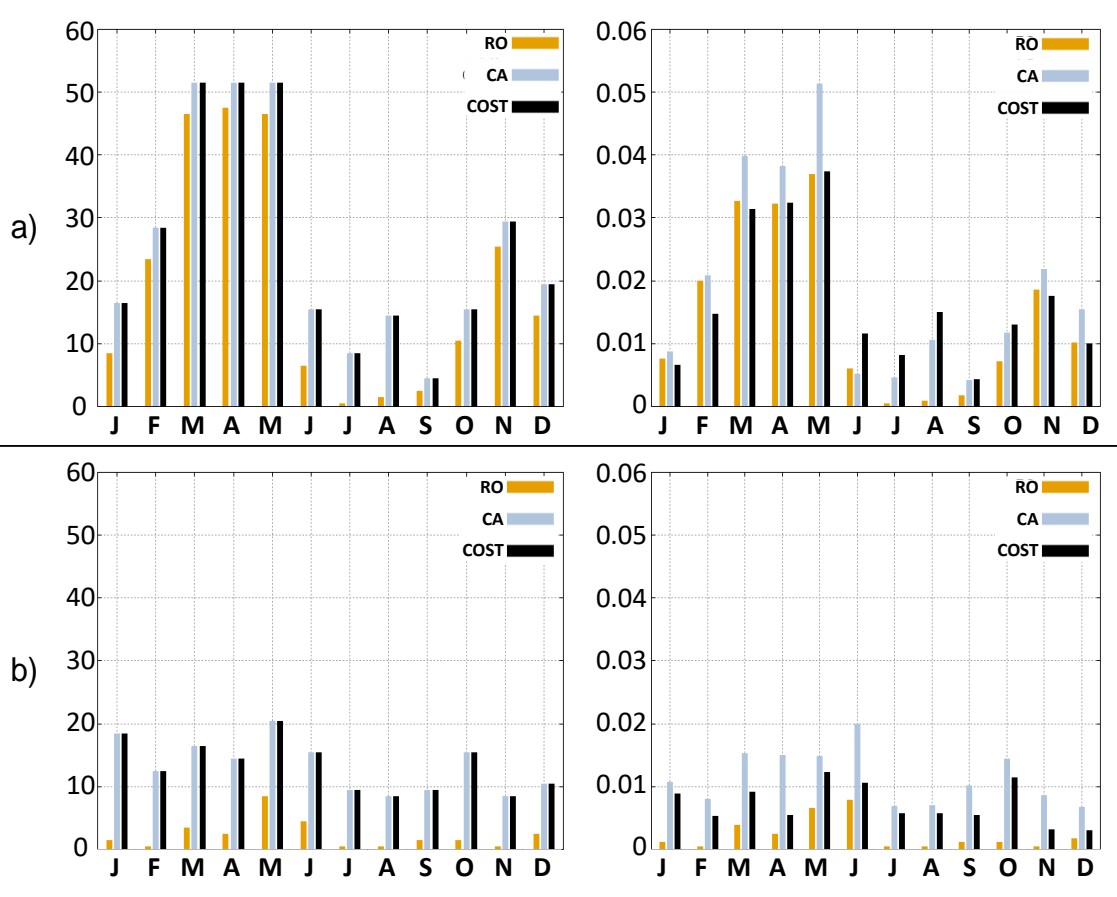

**Figure 6: Annual cycle of the values in the series associated with RO, CA and COST. Panel a** shows droughts, whereas **panel b** denotes extreme rainfall. The figures on the left represent the frequency (number of months with non-zero values), and those on the right depict normalised intensity. Intensity is normalised by dividing each datum by the maximum value of each series.

### 4.2 Seasonal variability and the coherence of hydrometeorological extremes

In seasonal and annual drought variability terms (see Figure 7), the three reconstruction methods coincide in pointing out the same long drought periods. However, the years with more intensified droughts differ. While for RO the 5 driest years were 1683, 1605, 1655, 1756 and 1606 (where the fifth position is shared with the years 1749, 1765 and 1774), for CA this ranking includes the years 1627, 1606, 1628, 1660 and 1756. Finally for COST, the five driest years were 1756, 1627, 1879, 1689 and 1628, respectively. In this way, a clear intermethod agreement is reached in that the years 1606, 1627, 1628, and, especially 1756, were the driest. These droughts are consistent with the results of other works. Corona et al. (1988) detected a severe drought between 1602 and 1606 to the SW of the Iberian Peninsula based on dendroclimatic evidence. Martín-Vide and Barriendos (1995: 212) identified 1628 as one of the four driest years in Barcelona for the 1525-1825 period. The year 1756 witnessed wide climate variability in the Iberian Peninsula, characterised by severe droughts (Cuadrat et al., 2016: 72), and aggravated by torrential rains and a severe locust plague (Alberola Romá, 1996).

The seasonal drought values in each year also show wide temporal variability, as well as notable differences between methods. The agricultural model of Caravaca, based on cereal production, generally magnifies the impact of spring droughts. Winters are, however, less sensitive, but were especially dry between 1600 and 1615. The autumns of the second half of the 17th and 18th centuries increased the relative weight of droughts in annual computations. Although the Caravaca society has adapted to lack of precipitation in summer, COST, and CA to a lesser extent, indicate very dry summers in the 17th century and in the second half of the 19th century. However, RO is unable to provide any information about droughts for these months because RO ceremonies are lacking.

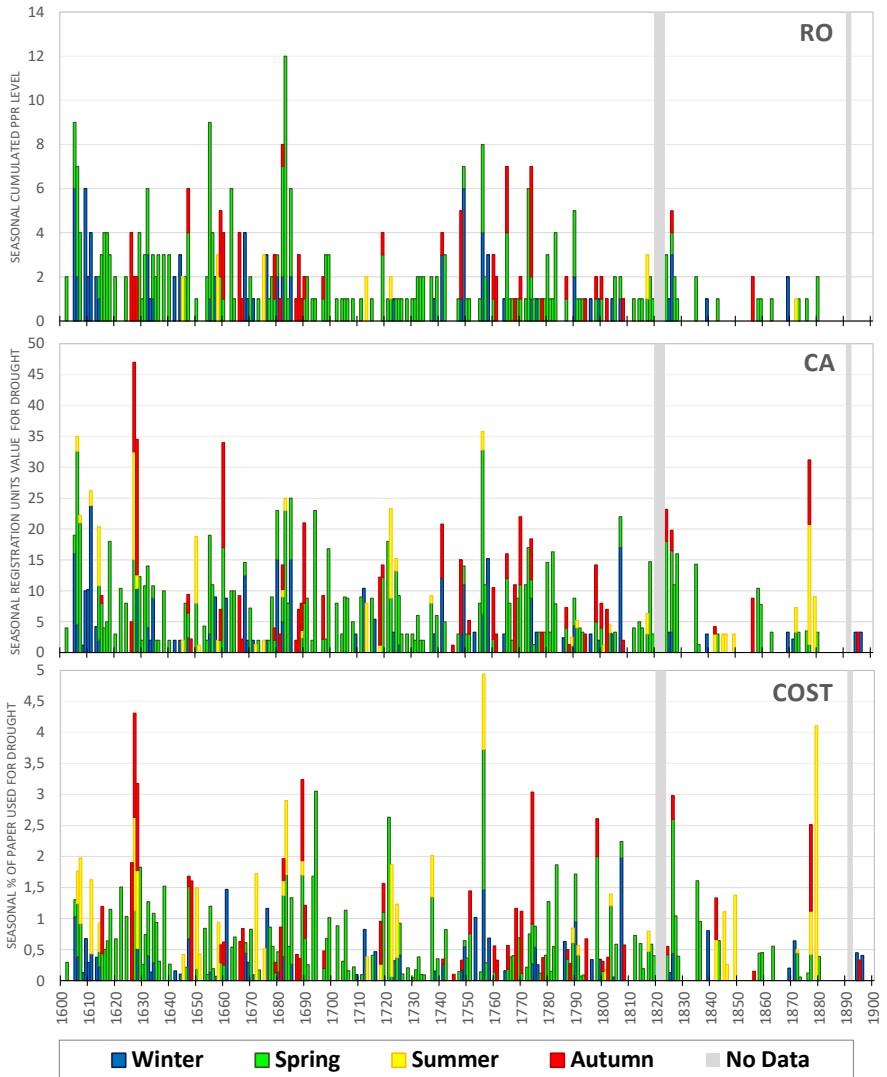

**Figure 7: Seasonal drought variability in Caravaca between 1600 and 1900.** Seasonal values accumulate for each year. In this way, the annual value is defined as the sum of the four seasonal values for a given year. The values on the Y axis are in the unit that corresponds to each methodology; i.e. the sum of the Pro-Pluvia RO levels for each station in RO, the sum of the values of RU for the drought in CA, and the percentage of paper used for droughts during each season in COST. *Data gaps (from 1820 to 1823, from 1891 to 1892) are shown in grey.

The annual and seasonal series of indices associated with extreme rainfall offer some interesting aspects (Figure 8). The rarity of extreme rainfall reconstructed by RO is remarkable as only 4 months had extreme rainfall events throughout the 19[th] century. This situation reduces this method's reliability to reconstruct precipitation. Note that this factor does not affect the CA and COST methods, which were able to record many extreme rainfall events throughout the study period. Both CA and COST report wet winters at the end of the 17[th] century. This period is consistent with the coldest and wettest winters detected for the same period by Alcoforado et al. (2000) to the south of the Iberian Peninsula, and might indicate a strengthening of the zonal circulation associated with the polar front during the late Maunder Minimum (Luterbacher et al., 2001), which is also consistent with climate simulations (Gómez- Navarro, et al., 2011). In the second half of the 18[th] century, especially between 1855 and 1870, several wet winters are also detected, and coincide with wet winters in Andalusia (Sánchez-Rodrigo et al., 2000). Spring shows generally more homogeneous behaviour, and no obvious anomalous periods. In autumn and summer, notable differences appear between CA and COST. CA tends to exaggerate the intensity of several humid summers and autumns, especially in the 17[th] century. Both methods coincide in indicating the autumns of 1740-1775 as being very humid, like the autumns in the second half of the 19[th] century. Barriendos and Llasat (2009) highlight severe autumn rain in the second half of the 19[th] century, and associate it with a period of strong climate variability to the east of the Iberian Peninsula, identified as the Maldá Anomaly. There are also many historic studies on and testimonies from this epoch[5] that point out the sharp intensity of autumn rainfall in the Spanish Mediterranean Region in the second half of the 19[th] century (Barriendos and Martín-Vide, 1998; Gil-Guirado, 2013) (Figure 8). In line with all this, an excellent source of evidence for rain intensity during this period is pointed out by the fact that five of the ten rainiest years detected by CA and COST are included in the 1829-1888 period, with three of these years only between 1650 and 1685.

---

[5] In this climate context, and after the notable damages caused by the continuous floods in the late 19[th] century in SEf Spain, a number of studies were conducted to mitigate the negative impacts of floods. Indeed the work of Hernandez Amores (1885) entitled: "*Inundaciones de la huerta de Murcia: juicio sobre su frecuente repetición de pocos años a esta parte, sus terribles desastres, sus causas y remedios*" is a prominent example.

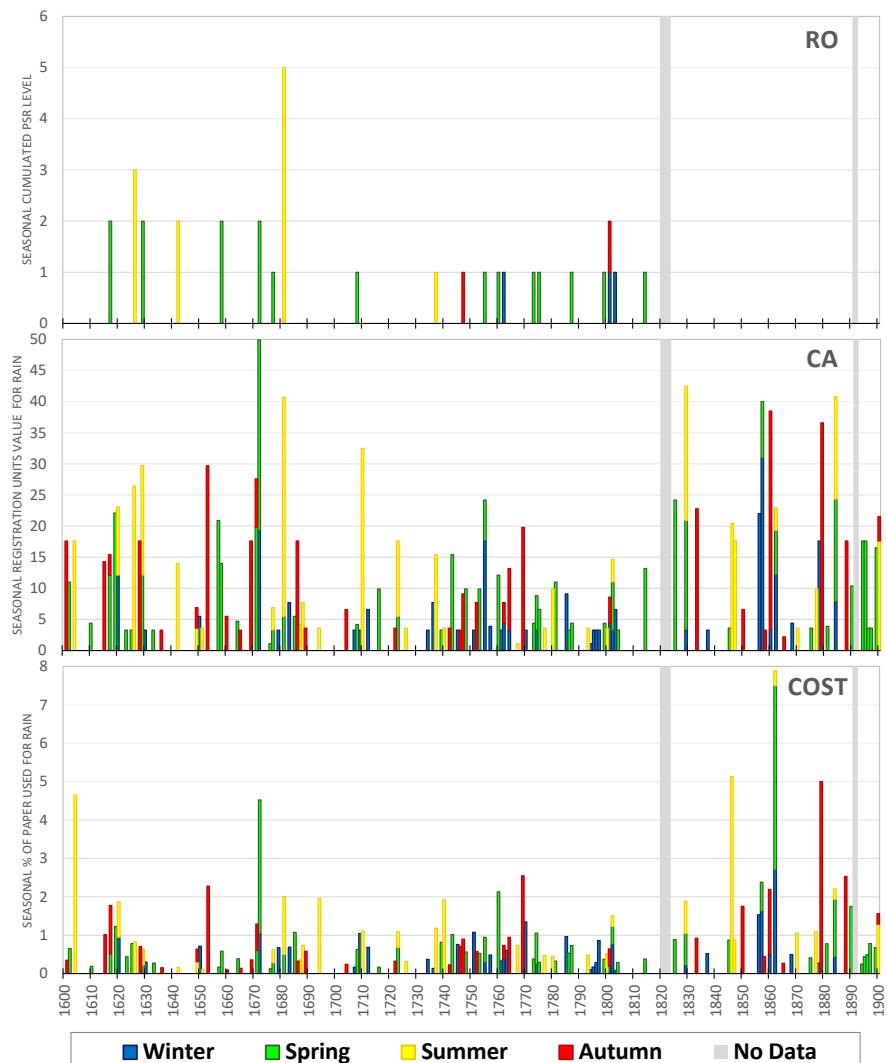

**Figure 8: Seasonal extreme rainfall in Caravaca between 1600 and 1900.** Seasonal values accumulate for each year. In this way, the annual value is defined as the sum of the four seasonal values for a given year. The values on the Y axis are in the unit that corresponds to each methodology, i.e. the sum of the Pro-Serenitate Rogations levels for each station in RO, the sum of the values of RU for drought in CA, and the percentage of paper used for drought during each season in COST. *Data gaps (from 1820 to 1823 and 1891 to 1892) are shown in grey.

To gain insight into the intra-annual differences of variability across the three methods, we need to bear in mind several considerations. Regarding droughts, it is important to take into account their continuous nature. Droughts show their social impacts cumulatively over time, which makes it difficult to define clear boundaries (Logar, & van den Bergh, 2013). This can be seen in our results, which account for the number of dry summers and were immediately preceded by a dry spring (see Table 2). More than 50% of summers with drought were preceded by a dry spring. However, these correlative droughts do not occur as frequently in winter, spring or autumn. This allows us to infer further information about the severity of spring drought when both CA and COST indicate dry summers. A similar analysis reveals clear differences for extreme

rainfall events. Unlike droughts, extreme rainfall manifests its impact immediately and, therefore, documents reflect them with no delay or with references to previous events. This explains the lack of autocorrelation noted between consecutive seasons. Both CA and COST detect a correlation between wet summers and previous wet springs (see Table 2).

| | CA and COST | RO |
|---|---|---|
| **Drought** | | |
| Dry summer preceded by dry spring | 54.55% | 28.57% |
| Dry spring preceded by dry winter | 13.95% | 12.61% |
| Dry autumn preceded by dry summer | 15.91% | 0.00% |
| Dry winter preceded by dry autumn | 16.98% | 12.50% |
| **Extreme rainfall** | | |
| Rainy summer preceded by rainy spring | 25.81% | 0.00% |
| Rainy spring preceded by rainy winter | 16.00% | 0.00% |
| Rainy autumn preceded by rainy summer | 6.45% | 0.00% |
| Rainy winter preceded by rainy autumn | 10.81% | 0.00% |

**Table 2: Temporal coincidence of drought and extreme rainfall values between consecutive seasons.** Each cell shows the percentage of dry/rainy seasons that were preceded by dry/rainy seasons. *Different colours symbolise how much the values deviate above (from white to red) or below (from blue to white) from the 50th percentile of the data, respectively.

### 4.3 Breakpoints and series consistency

For the drought data, the Pettitt, SNHT and Buishand tests identify breakpoints in both intensity and frequency (see Table 3). For RO and COST, the Pettit and Buishand tests identify reduced drought intensity towards the end of the 17th century. The Buishand test points out that 1783 was the year with marked changes towards lesser drought severity conditions. However, all three methods agree that 1828 was the year when an abrupt change towards less intense, and especially less frequent, droughts took place. We attempted to determine whether this jump is related to an actual change in climate conditions as opposed to a methodological artifact. It is important to bear in mind that during this period, Spain underwent important social changes and continuous conflicts between public authorities, so we should act cautiously to not absolutely rule out the possibility of the detected breakpoints being related to consistency issues in documentary sources. The joint data analysis and the existing scientific literature discussed in the previous section are consistent with the reduction in droughts from the first third of the 19th century (Barriendos and Martín-Vide, 1998). The Buishand test also detects a remarkable increase in extreme rainfall intensity for the COST method after 1844. This increased rainfall intensity is consistent with the more abundant rainfall detected in the literature (Sánchez Rodrigo y Barriendos, 2008; Barriendos and Llasat, 2009; Benito, Rico et al., 2010). All these independent sources of insight indicate that the breakpoints identified by tests are not due to documentary sources. Instead the detected changes are climate-related in nature, which validates the methodology implemented to reconstruct droughts, and renders the COST approach as being the most consistent to analyse extreme rainfall variability.

| | | Intensity | | | | Frequency | | | |
|---|---|---|---|---|---|---|---|---|---|
| | | $T$ | *p-value* | *1st $\overline{X}$** | *2nd $\overline{X}$** | $T$ | *p-value* | *1st $\overline{X}$** | *2nd $\overline{X}$** |
| | | **Pettit Test** | | | | | | | |
| **RO** | Drought | **1685** | 0.0012 | 2.349 | 0.943 | **1828** | < 0.0001 | 0.689 | 0.171 |
| | Rain | 1681 | 0.2408 | | | 1814 | 0.0900 | | |
| **CA** | Drought | **1828** | 0.0102 | 6.862 | 1.974 | **1828** | < 0.0001 | 0.756 | 0.329 |
| | Rain | 1855 | 0.2002 | | | 1893 | 0.3477 | | |
| **COST** | Drought | 1694 | 0.0654 | | | **1828** | < 0.0001 | 0.756 | 0.329 |
| | Rain | 1844 | 0.1003 | | | 1893 | 0.3812 | | |
| | | **Buishand Range** | | | | | | | |
| **RO** | Drought | **1783** | < 0.0001 | 1.837 | 0.550 | **1828** | < 0.0001 | 0.689 | 0.171 |
| | Rain | 1681 | 0.0655 | | | **1814** | 0.0430 | 0.102 | 0 |
| **CA** | Drought | **1783** | < 0.0001 | 7.204 | 3.212 | **1828** | < 0.0001 | 0.756 | 0.329 |
| | Rain | 1855 | 0.2260 | | | 1733 | 0.2744 | | |
| **COST** | Drought | **1694** | 0.0062 | 0.780 | 0.446 | **1828** | < 0.0001 | 0.756 | 0.329 |
| | Rain | **1844** | 0.0248 | 0.328 | 0,799 | 1733 | 0.2744 | | |
| | | **SNHT** | | | | | | | |
| **RO** | Drought | **1828** | < 0.0001 | 1.702 | 0.229 | **1828** | < 0.0001 | 0.689 | 0.171 |
| | Rain | 1814 | 0.0992 | | | 1814 | 0.0900 | | |
| **CA** | Drought | **1828** | < 0.0001 | 6.862 | 1.974 | **1828** | < 0.0001 | 0.756 | 0.329 |
| | Rain | 1735 | 0.9766 | | | 1733 | 0.5618 | | |
| **COST** | Drought | **1828** | < 0.0001 | 0.639 | 0.278 | **1828** | < 0.0001 | 0.756 | 0.329 |
| | Rain | 1735 | 0.9766 | | | 1733 | 0.5618 | | |

**Table 3: Results of the homogenisation tests for breakpoint detection.**
T refers to the year in which an lack of data homogeneity is detected. To calculate the p-value, 10,000 Monte Carlo simulations were run. The years when the p-value is below the alpha=0.05 level of confidence are highlighted in bold.
* The 1st $\overline{X}$ refers to the average over the period prior the breakpoint, while the 2nd $\overline{X}$ corresponds to the average for the period after it.

## 4.4 Secular variability and the temporal space coherence of reconstructions

As discussed in the previous section, all the reconstruction methods indicate a significant decline in the number and severity of droughts at the end of the study period (Figure 9). This decrease is significant (95% level with the Mand-Kendall test) in the summer, autumn and annual series, and also in terms of both frequency and intensity. CA and COST identify a concurrent increase in extreme rainfall during this period, which is climatically consistent with reduced drought severity, while RO depicts reduced extreme rainfall. Regarding this difference to the other two methods, it is noteworthy that the last PSR was held in 1814, which suggests loss of homogeneity of documentary sources, and therefore questions the ability of RO to reconstruct extreme rainfall with these data.

We note that the timing of droughts and extreme rainfall in these series agrees qualitatively with independent palaeoclimatic evidence in locations close to the study area (Barriendos, 1996-1997, Barriendos, 1997, Zamora Pastor, 2002, Sanchez Rodrigo & Barriendos, 2008 Benito, Rico et al., 2010, Machado, et al., 2011). For droughts, the three methods offer high secular covariability, which highlights the consistency of the three methods to reconstruct the low frequency variability of this variable. RO exhibits decoupling periods according to CA and COST, especially at the end of the series. The periodicity of droughts presents cycles of approximately 30 years (see Figure 9: $a_1$ and $a_2$). Moreira et al., (2012) detected a drought periodicity of between 26 and 30 years in Portugal, and Chen et al., (2006) related this hydrological variability with the Sunspot Number variability with an almost 30-year periodicity. The differences for extreme rainfall among methods are

more evident, which is partly due to the scarcity of RO data from the 19[th] century. Although CA and COST are coherent during wet periods, differences are more marked than for droughts (see Figure 9: $b_1$ and $b_2$). CA tends to exaggerate the intensity of wet periods than COST, which can be attributed to the fact that historic documents tend to overstate the description of the effects of heavy rains because it was customary to apply tax reductions as compensation measures for catastrophes, which would therefore introduce a non-climate bias into documentary evidence (Gil-Guirado, 2013). Conversely, COST is not affected by this subjective bias and, therefore, becomes a more objective methodology.

The driest period in the 17[th] century went from 1620 to 1630, as also reported by dendroclimatic evidence in SE Spain (Creus Novau & Saz Sánchez, 2008). The following dry phase took place in the last quarter of the same century, during the Maunder Minimum, and is also reported in the bibliography (Barriendos, 1997, Creus Novau & Saz Sánchez, 2008). The generally low temperatures during this period might be related to a drop in the Mediterranean Sea temperature, which is the main contributor of moisture to autumn rains in the eastern Iberian Peninsula (Luterbacher et al., 2001). The 18[th] century is generally less dry than the previous century, but presents two important dry periods: the first around the decade of 1720, and the second between 1755 and 1785. Some authors have identified a cold phase in the first decades of this century (Mann et al., 2000) as a driver of the northern displacement of storm fronts associated with both the polar front and the western flow which, in turn, can be related to a positive phase of the NAO index (Creus Novau & Saz Sánchez, 2008, Sánchez Rodrigo and Barriendos, 2008). The 19[th] century, however, is the least dry of the whole study period. Finally, there are two short arid periods from 1825 to 1830 and from 1875 to 1880.

A heavy rainfall period occurred between 1750 and 1800, during the so-called Maldá anomaly, characterised by the consecutive occurrence of drought episodes, followed by extraordinary floods with catastrophic consequences (Barriendos & Llasat, 2009; 2010). Of all the humid periods, that which took place in the second half of the 19[th] century stands out. It can be described as the wettest of the study period, and is a distinct feature of the final phase of the Little Ice Age in the Mediterranean lands of the Iberian Peninsula (Benito, Rico et al., 2010, Machado, Benito, Barriendos & Rodrigo, 2011, Zamora Pastor, 2002, Sánchez Rodrigo & Barriendos, 2008).

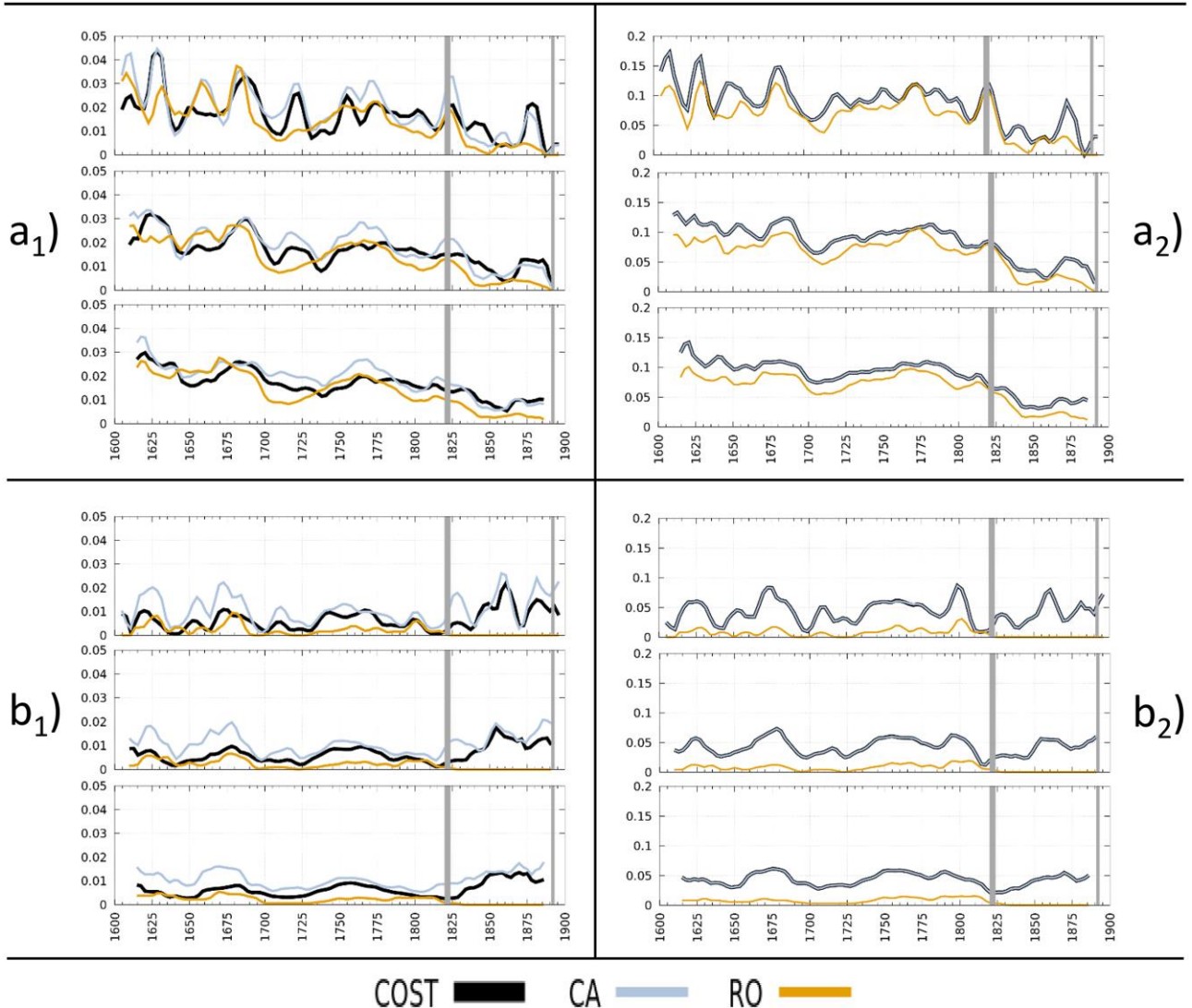

**Figure 9: Normalised intensity and occurrence of droughts and extreme rainfall in Caravaca between 1600 and 1900.** Series are normalised by dividing each datum by the maximum value of each series. Intensity is defined as the normalised monthly value, while occurrence is defined dichotomously by differentiating between the months when an event occurred, 1, from the months with no event, 0. **Panel a₁** shows the running mean of drought intensity, while **Panel a₂** shows the running mean of droughts occurring. **Panels b₁** and **b₂** depict the same information, but for extreme rainfall reconstruction. All the panels show the running mean with a temporal window of 10, 20 and 30 years, respectively. *Data gaps (from 1820 to 1823 and 1891 to 1892) are shown in grey.

## 5 Conclusions

HC has some methodological limitations that make it difficult to produce climate reconstructions in sparsely populated areas, which leads to a major geographic bias in past climate variability reconstruction. The richness of worldwide documentary heritage calls for new methodologies to be developed to overcome such methodological limitations by allowing new high-

resolution multisecular reconstructions to be done. This paper proposes a new methodology (the COST method) and compares it with a new approach, a CA. Both methodologies are identified as robust alternatives to a more classic method consisting in the analysis of RO. One improvement of CA and COST, compared to RO, is that the first two are capable of capturing more drought events than RO. Yet the most significant novelty of these approaches is that they allow events

associated with rainfall to be reliably reconstructed, while classic methods are limited to reconstruct situations associated with water deficit. This work also demonstrates that the implemented methods enable reconstructions to be carried out on a monthly basis. Indeed the validation of the robustness of monthly results can be made by comparing the output of reconstructions with the observed annual rainfall cycle and the progress made by agricultural work in the study area.

These three approaches are applied exemplarily to produce the reconstruction of hydrometeorological extremes in a small

town in southern Spain, namely Caravaca. The three proposed methods identify reduction in droughts to the SE of the Iberian Peninsula, which is compatible with the results of other studies. However, COST is the only method capable of detecting a change towards more intense rains from the mid-19[th] century. This fact indicates that such trends are due to actual climate variability, rather than to data non-homogeneities. All the methods detect anomalous key periods during the Little Ice Age that mirror those reported in previous studies. That is, a dry phase takes place in the fourth quarter of the 17[th]

century during the Maunder Minimum, and is related to a drop in the Mediterranean Sea temperature. Another dry phase occurs around the decade of 1720, and a second one between 1755 and 1785, which can be related with a positive phase of the NAO index. CA and COST also detected a number of wet phases, and a heavy rainfall period occurred between 1750 and 1800 during the so-called Maldá anomaly. From all the humid periods, that which took place in the second half of the 19[th] century stands out. This period can be described as the wettest one during the study period, and is a distinct feature of the

final Little Ice Age phase in Mediterranean lands of the Iberian Peninsula.

HC techniques are often affected by lack of objectivity, which remains a problem and one that needs to be contemplated when assessing the reliability of reconstructions. The COST method is a significant step forward in achieving a more objective approach to analyse historic documents. This method bases its implementation on an economic variable by, therefore, limiting biases associated with non-climate matters. So while RO and CA can overstate situations of excess water

or water deficits due to non-climate motivations, i.e. paying fewer taxes as a compensation measure, COST values present greater homogeneity and are more closely related to the climate reality of the time. As COST is based on the amount of discourse around a climate variable, it enables comparative exercises to be made in regions with different languages and ways of understanding the natural environment. Finally, COST has the advantage of being a simpler approach to decode documentary information than RO and CA.

In this way, CA, and especially COST, allow currently underexploited documentary evidence to be used by multiplying the potential number of regions where historic techniques can be used to produce new climate reconstructions. We believe that this may open up new fieldwork opportunities to motivate follow-up research work.

## Appendix A

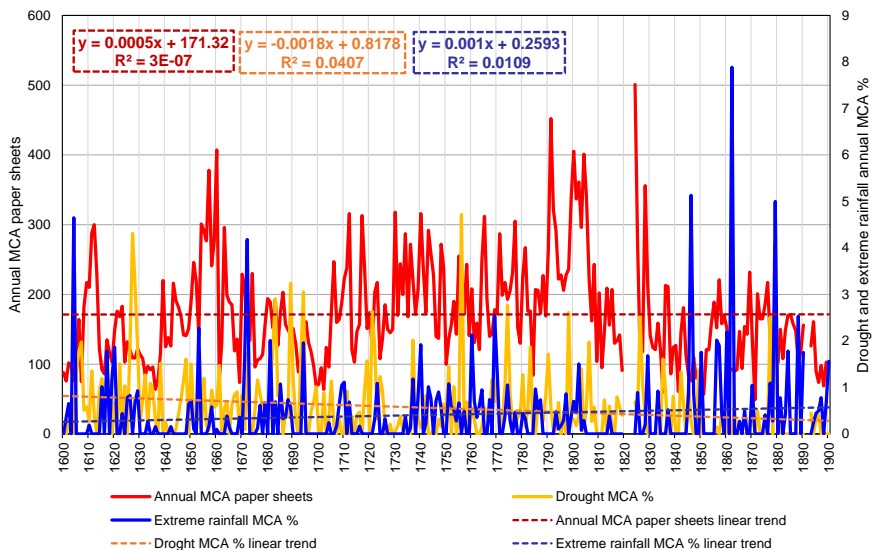

**Figure A1:** Annual MCA paper Sheets and Sheets percentage used for drought and extreme rainfall in Caravaca (1600-1900).

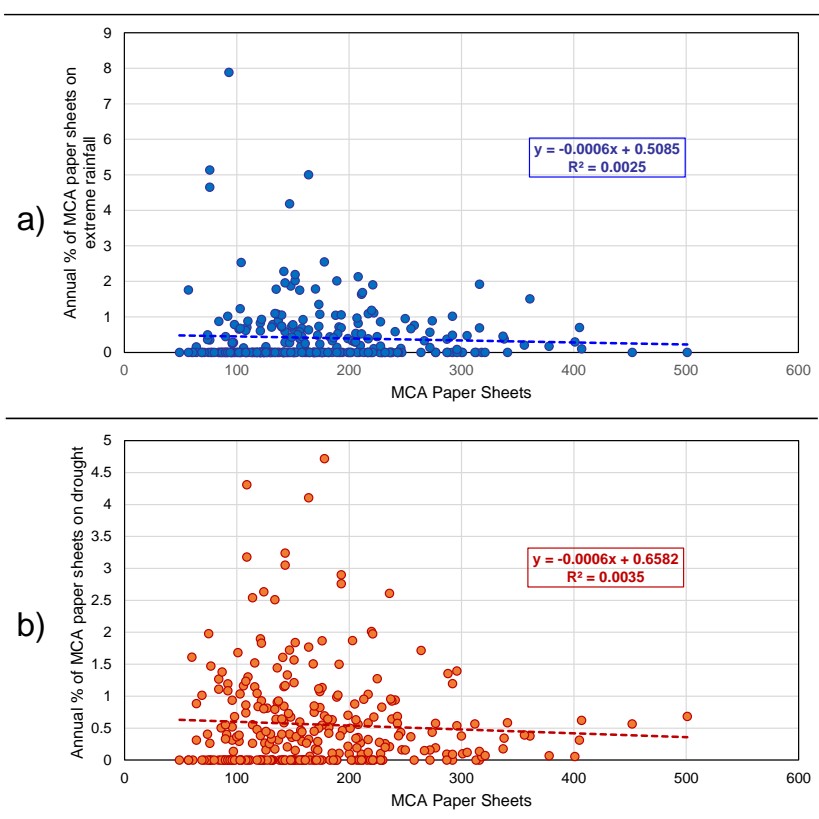

5    **Figure A2:** Relationship between the annual amount of MCA paper sheets and the annual percentage of MCA paper sheets dedicated to inform about droughts (Panel a) and extreme rainfall (Panel b).

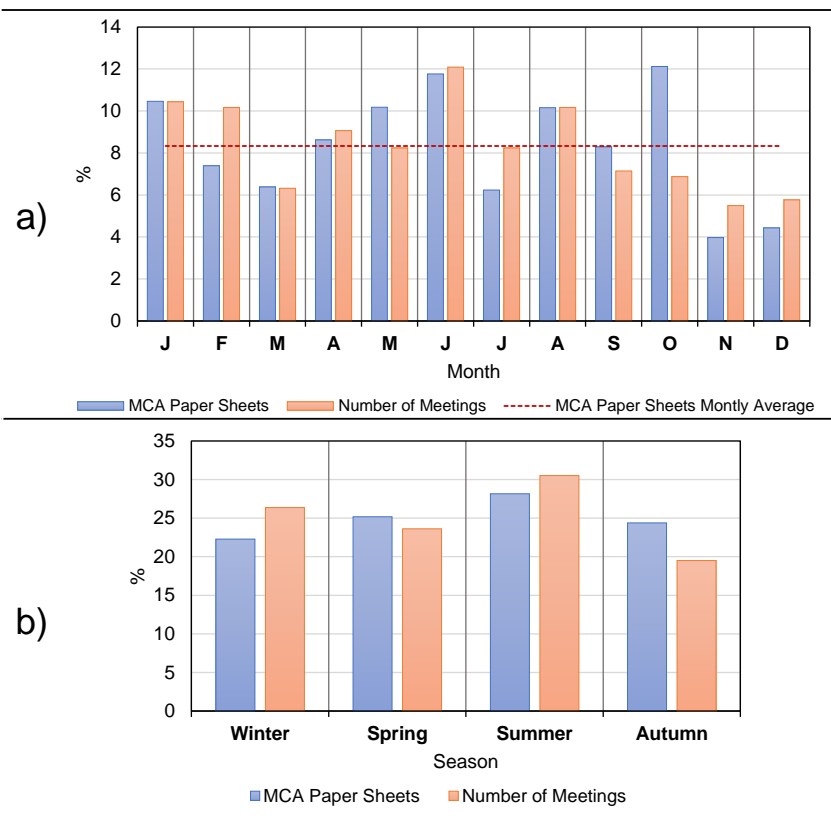

**Figure A3:** Amount of MCA paper sheets used by month (Panel a) and season (Panel b) in Caravaca. The values are the averages for the years 1698, 1614, 1657, 1749, 1800, 1850 and 1983.

**Appendix B (acronyms used throughout the manuscript):**

**COST:** Cost Opportunity for Small Towns

**RO:** Rogations Method

**CA:** Content Analysis

**WMO:** World Meteorological Organization

**HC:** Historical Climatology

**CC:** Capital City

**EP:** Episcopal sees

**MCA:** Municipal Chapter Acts

**CCO:** City Council

**RU:** Registration Units

**PPR:** pro-pluvia rogations

**PSR:** pro-serenitate rogations

*Acknowledgments:* S.G.G. acknowledges the support of the Spanish Ministry of Science, Innovation and Universities through a "Juan de la Cierva-Incorporación" grant (IJCI-2016-29016).

J.J.G.N. acknowledges the CARM for the funding provided both through the Seneca Foundation (Project 20022/SF/16), as well as the "Juan de la Cierva-Incorporación" programme (IJCI-2015-26914).

*Competing interests.* The authors declare that they have no conflict of interest.

*Data availability.* The systematic data are not publicly accessible because they are currently being used in an ongoing research project. However, the research data required to reproduce the work can be obtained upon request by contacting the Corresponding Author (salvagil.guirado@ua.es).

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
