# Peer review of "The weather behind words. New methodologies for integrated hydrometeorological reconstruction through documentary sources."

_Climate of the Past, 2019_

## Referee Comment (RC1) · Anonymous Referee #1 · 15 Feb 2019

Guil-Guirado et al. present in this manuscript a new and interesting methodology to infer climatic conditions in the past through historical documents. In addition, they compare it with the most commonly used methodologies to infer past climate variability with historical documents, including appealing results. The manuscript is consistent and the results are robust, although I suggest some changes to facilitate its reading and compression before being published. In addition, the manuscript needs a thorough revision of the language. Shortening the sentences and making a section with all the acronyms would benefit the manuscript comprehension too.

[Figure]

Right now the methodology is rather confusing. I would suggest clarifying it by doing flow diagrams for each of the proposed methodologies (Page 8). Then, it would be very convenient to make an annex or table with the acronyms. Another important point is to include the whole methodology in the corresponding section. Sections 4.3 and 4.4 contain information on the methodology which should be included, for example, as section 3.4. Climate analysis of drought series.

Some minor comments; Line 29/1. Define and cite WMO. Line 6/2. Please rephrase. Line 25/2. You might want to replace further by improve. Line 9/9. have promoted.. Line 15/14. La? Table 1 (Page14). To improve the visibility, I would suggest to plot it in columns following the color code of the next figures. Lines 1-5/22. What are the potential climatic implications of such cycles? e.g. Solar irradiation? Line 18. period IN the.. Line 8/23. Results section again? I guess it should be discussion instead. Please be attentive to all details before submitting a manuscript.

---

## Referee Comment (RC2) · Anonymous Referee #2 · 6 Mar 2019

1. Overall comment This paper is very interesting and brings a new insight on methodological approaches of historical climate reconstructions based on documentary data. The authors apply a new method of hydrometeorological reconstruction and attempt its validation by applying it simultaneously with other two already known methods. The manuscript is well designed and properly supported on up to date bibliography. However, this version exhibits some problems that must be overcome, and some questions should be attentively considered in order to improve the quality and robustness of the work.

Main comments 1) As the most innovative contribution of this paper is the method named "COST", some important details are missing in its description. For instance, along the study period (1600-1900) the "Actas Capitulares" (AC) of the city council have maintained the same model, that means, the same periodicity, structure and general dimension? If the answer is positive, so clarify by stating it. Note that a single example of AC is shown but the reader has no information if this sample is valid over the three centuries. Does the frequency of the council meetings is satisfactorily suitable to permit the data collection with monthly resolution? The authors only mention the total number of consulted sheets of paper but did not make any reference to the number of books and municipal chapter acts and it's interannual distribution. In my opinion this must be clearly justified, because it concerns the consistency of the study. 2) Regarding the methodology, there are several important details that should be clearly stated in the text instead of being included in the tables and figures captions. The reading and comprehension of paper is difficulted by this fact, in my opinion. I suggest an improvement of section 3, providing a more clear and detailed description of all methodological procedures undertaken through the study. 3) There are some important problems detected in Figures as follows: a) Title of Figure 7 (drought variability) is not suitable e should be modified according the Figure 8 title (extreme rainfall variability). b) Figure 9 is not legible and must be resized.

Minor comments. 1) The text needs a general revision of the English. There are several unclear expressions, some mistakes and missing words. I suggest a general revision of the text redaction. 2) The titles of sections 2 and 3 should be modified because the authors should point out the sources and methods used in their own study and not in such general mode as "Sources in Historical Climatology". In my opinion this is incorrect. 3) The final section must be a "Conclusion" instead of "Results" (repeated section title).

Please also note the supplement to this comment:
https://www.clim-past-discuss.net/cp-2019-1/cp-2019-1-RC2-supplement.pdf

---

## Author Comment (AC1) · 2 Apr 2019

**Referee #1 responses:**

**Overall comment:**

− **RC**: Gil-Guirado et al. present in this manuscript a new and interesting methodology to infer climatic conditions in the past through historical documents. In addition, they compare it with the most commonly used methodologies to infer past climate variability with historical documents, including appealing results. The manuscript is consistent and the results are robust, although I suggest some changes to facilitate its reading and compression before being published. In addition, the manuscript needs a thorough revision of the language. Shortening the sentences and making a section with all the acronyms would benefit the manuscript comprehension too.

− **AC:** Thank you for your flattering and constructive comments. Our aim was to offer new methodological approaches to historical climatology that can be suggestive to extend the knowledge of the historical climate evolution in regions that have been barely studied so far. We appreciate and understand your comments about the language review. Therefore, after taking care of all the suggestions by both reviewers, the full text has been carefully edited by a professional translator.

**Main comments:**

− **RC**: Right now the methodology is rather confusing. I would suggest clarifying it by doing flow diagrams for each of the proposed methodologies (Page 8). Then, it would be very convenient to make an annex or table with the acronyms. Another important point is to include the whole methodology in the corresponding section. Sections 4.3 and 4.4 contain information on the methodology which should be included, for example, as section 3.4. Climate analysis of drought series.

− **AR:** We mostly agree with these issues. For this reason, some parts of the work, especially the methodology, have been re-written to improve their clarity. In detail, section 3 (methodology) has been modified in order to include clarifications about the implementation of each method. In addition, we have included new flow diagrams for each of the proposed methodology as additional panels in figures 3, 4 and 5, respectively (see attached figures). Finally, we have included the methodological information contained in sections 4.3 and 4.4 as an additional subsection in the methodology section (3.4). Finally, and following the referee's recommendations, we have included an Annex (See Annex 2) where we compile all the acronyms used throughout the manuscript.

**Minor comments:**

− **RC**: Some minor comments; Line 29/1. Define and cite WMO. Line 6/2. Please rephrase. Line 25/2. You might want to replace further by improve. Line 9/9. have promoted. Line 15/14. La? Table 1 (Page14). To improve the visibility, I would suggest to plot it in columns following the color code of the next figures. Lines 1-5/22. What are the potential climatic implications of such cycles? e.g. Solar irradiation? Line 18. period IN the.. Line 8/23. Results section again? I guess it should be discussion instead. Please be attentive to all details before submitting a manuscript.

− **AR**: Thank you very much for highlighting these important details and providing advice about the convenience of including some necessary clarifications. All these issues have been taken into account. Regarding the potential of cycles of around 30 years detected in droughts, although the reason remains unclear, some authors (Moreira et al., 2012)

have detected a similar periodicity of droughts in Portugal. In addition, as you suggest, Chen et al., (2006) relate this hydrological variability with the Sunspot Number variability. We have added a short comment in the new version of the manuscript about this particular point.

Here is a more detailed description of the changes carried out:

− *Line 29/1. Define and cite WMO*. This is now defined
− *Line 6/2: This text* has been rewritten
− *Line 25/2* Done
− *Line 9/9:* Corrected
− *Line 15/14:* Corrected
− *Table 1 (Page14). To improve the visibility, I would suggest to plot it in columns following the color code of the next figures.* Following this advice, we have used the same color code of figure 6 to the same variables of table 1.
− *Lines 1-5/22. What are the potential climatic implications of such cycles? e.g. Solar irradiation?.* We have added some discussion about the detection of these cycles in Portugal, as well as the possible relationship between these cycles and solar variablity.
− *Line 18/22*. Corrected
− *Line 8/23.* Corrected and section renamed

[Figure]

**Figure 1:** RO method by step (Panel a) and encoding example of the RO method (Panel b). This particular example refers to a Pro-Pluvia RO on 18th April 1698, so the reconstructed variable is drought. Source: the Carmesi Project

[Figure]

**Figure 2:** Content Analysis method by step (Panel a) and an example of the encoding of the Content Analysis (CA) method (Panel b). The coded source is the same as in Figure 2 to emphasise how the three different approaches are applied in practice. This particular example refers to a PPR on 18 April 1698, so the reconstructed variable is drought. Source: the Carmesi Project.

[Figure]

**Figure 3:** COST method by step (Panel a) and an example the COST method encoding (Panel b). The coded source is the same as in Figure 2 and 3 to emphasise how the three different approaches are applied in practice. This particular example refers to a PPR on 18 April 1698, so the reconstructed variable is drought. Source: the Carmesi Project.

**ANNEX 2 (acronyms used throughout the manuscript):**

**COST:** Cost Opportunity for Small Towns

**RO:** Rogations Method

**CA:** Content Analysis

**WMO:** World Meteorological Organization

**HC:** Historical Climatology

**CC:** Capital City

**EP:** Episcopal sees

**MCA:** Municipal Chapter Acts

**CCO:** City Council

**RU:** Registration Units

**PPR:** pro-pluvia rogations

**PSR:** pro-serenitate rogations

---

## Author Comment (AC2) · 2 Apr 2019

**Referee #2 responses:**

**Overall comment:**

- RC: This paper is very interesting and brings a new insight on methodological approaches of historical climate reconstructions based on documentary data. The authors apply a new method of hydrometeorological reconstruction and attempt its validation by applying it simultaneously with other two already known methods. The manuscript is well designed and properly supported on up to date bibliography. However, this version exhibits some problems that must be overcome, and some questions should be attentively considered in order to improve the quality and robustness of the work.
- AR: Thank you for your kind considerations and for your very pertinent clarifications and qualifications. We appreciate these comments, which will undoubtedly increase the quality, reliability and robustness of our paper. We hope that after these changes are implemented, the proposed new methodology could become useful for future research.

**Main comments:**

- RC: As the most innovative contribution of this paper is the method named "COST", some important details are missing in its description. For instance, along the study period (1600-1900) the "Actas Capitulares" (AC) of the city council have maintained the same model, that means, the same periodicity, structure and general dimension? If the answer is positive, so clarify by stating it. Note that a single example of AC is shown but the reader has no information if this sample is valid over the three centuries. Does the frequency of the council meetings is satisfactorily suitable to permit the data collection with monthly resolution? The authors only mention the total number of consulted sheets of paper but did not make any reference to the number of books and municipal chapter acts and it's interannual distribution. In my opinion this must be clearly justified, because it concerns the consistency of the study.
- AR: Thank you for raising this important point, which is undoubtedly of great importance and must be clarified accordingly. Indeed, the AC have maintained the same structure and composition throughout the full studied period. We have shown this by adding an additional panel (Figure 1 below) with up to 6 examples of AC distributed throughout the full period. In these examples, we can observe how the structure and composition of the ACs have barely changed through time. This is so because the ACs were official documentation that had to be endorsed by official state paper. As such, the government required (and it was mandatory by law) that the structure should be the same, and that it should be consistent through time.

---

## Author Response (AR1)

Many thanks to the two referees and editor for their detailed and constructive reports.

We have made the changes in the original manuscript as reflected in the previous responses to the reviewers. In the revised version of the manuscript, all the changes in relation to the comments of the reviewers and editor are marked in yellow and with active change control. Additionally, a thorough revision of the language has been carried out by a professional translator native in English. In order to facilitate the monitoring of the changes, we have decided not to mark the language corrections with control changes. However, if you consider that it is necessary to attach a version where the changes in the language are reflected with change control, we will be happy to prepare such a document.

Additionally to the two anonymous reviewers, we have considered all your recommendations. In this regard, in the introduction we have included the references that you have recommended, together with other additional references that have updated the bibliography used. We have also translated the Spanish text of figures 2, 3 and 4, including an approximate translation (the original text is written in old Castilian, which makes the translation more difficult and approximate) in the figure caption of these figures. We have also changed the title of section 5, which is now called Conclusions (sorry for the mistake). We have also corrected the expression "la table" (again, apologies for the mistake).

Other mayor changes have led to the inclusion of two Appendices. In Appendix A there are three figures that reinforce the feasibility of using the percentage of sealed paper as a proxy to reconstruct droughts and extreme rainfall through the COST method. In section 3.3 the changes to reinforce this hypothesis are marked in yellow. Appendix B is related to the consistency in the use of acronyms. Following the recommendations of the reviewers, in Appendix B we list all the acronyms used in the work and have made changes to the work so that this nomenclature is consistent. Additionally, we introduced changes in the nomenclature of the reconstructed variables to be consistent in the definition of the extreme rainfall reconstructed variable. Finally, we have modified figures 2, 3, 4, 5 and 9, as well as Table 1 to include the recommendations of both reviewers.

Other small style changes have been included to improve the readability of the manuscript.

In summary, all the recommendations made by you and the reviewers have been considered, making the changes in the manuscript in relation to these recommendations and suggestions. In this way, we attach the revised manuscript with control changes to reflect the changes in relation to the comments of the reviewers and editor.

Please, if you need any clarification, information or additional document, do not hesitate to contact us.

Below we reflect the specific changes in the manuscript in relation to the suggestions of both reviewers:

**Referee #1 responses:**

**Main comments:**

- **RC**: Right now the methodology is rather confusing. I would suggest clarifying it by doing flow diagrams for each of the proposed methodologies (Page 8). Then, it would be very convenient to make an annex or table with the acronyms. Another important point is to include the whole methodology in the corresponding section. Sections 4.3 and 4.4 contain information on the methodology which should be included, for example, as section 3.4. Climate analysis of drought series.

- **AR:** Some parts of the work, especially the methodology, have been re-written to improve their clarity. In detail, section 3 (methodology) has been modified in order to include clarifications about the implementation of each method (page 9 lines 10 to 18 for the Rogations method; page 10 lines 8 to 11, page 11 lines 1 to 3 and lines 23 to 27, and page 12 lines 1 and 2, and lines 14 to 16 for the Content Analysis method; page 14 lines 22 to 35 and page 13 lines 4 to 7 and lines 12 to 19 for the COST method). In addition, we have included new flow diagrams for each of the proposed methods as additional panels in figures 3, 4 and 5, respectively. Finally, we have included the methodological information contained in sections 4.3 and 4.4 as an additional subsection in the

methodology section (3.4). Finally, and following the referee's recommendations, we have included an Appendix (See Appendix B) where we compile all the acronyms used throughout the manuscript.

**Minor comments:**

5 − **RC**: Some minor comments; Line 29/1. Define and cite WMO. Line 6/2. Please rephrase. Line 25/2. You might want to replace further by improve. Line 9/9. have promoted. Line 15/14. La? Table 1 (Page14). To improve the visibility, I would suggest to plot it in columns following the color code of the next figures. Lines 1-5/22. What are the potential climatic implications of such cycles? e.g. Solar irradiation? Line 18. period IN the.. Line 8/23. Results section again? I guess it should be 10 discussion instead. Please be attentive to all details before submitting a manuscript.

− **AR**: Thank you very much for highlighting these important details and providing advice about the convenience of including some necessary clarifications. All these issues have been taken into account. Regarding the potential of cycles of around 30 years detected in droughts, although the reason remains unclear, some authors (Moreira et al., 2012) have detected a similar periodicity of 15 droughts in Portugal. In addition, as you suggest, Chen et al., (2006) relate this hydrological variability with the Sunspot Number variability. We have added a short comment in the new version of the manuscript about this particular point.

Here is a more detailed description of the changes carried out:

− *Line 29/1. Define and cite WMO.* This is now defined in page 1 line 30.
20 − *Line 6/2: This text* has been rewritten (See page 2 lines 6 and 7).
− *Line 25/2 This text* has been rewritten (See page 2 lines 23 to 25).
− *Line 9/9:* Corrected
− *Line 15/14:* Corrected
− *Table 1.* Following this advice, we have used the same color code of figures 6 and 9 to the same 25 variables of table 1.
− *Lines 1-5/22.* We have added some discussion about the detection of these cycles in Portugal, as well as the possible relationship between these cycles and solar variability (See Page 25 lines 17 to 22).
− *Line 18/22.* Corrected
30 − *Line 8/23.* Corrected (section 5 renamed as Conclusions)

**Referee #2 responses:**

**Main comments:**

− **RC**: As the most innovative contribution of this paper is the method named "COST", some important details are missing in its description. For instance, along the study period (1600-1900) 35 the "Actas Capitulares" (AC) of the city council have maintained the same model, that means, the same periodicity, structure and general dimension? If the answer is positive, so clarify by stating it. Note that a single example of AC is shown but the reader has no information if this sample is valid over the three centuries. Does the frequency of the council meetings is satisfactorily suitable to permit the data collection with monthly resolution? The authors only mention the total number of 40 consulted sheets of paper but did not make any reference to the number of books and municipal chapter acts and it´s interannual distribution. In my opinion this must be clearly justified, because it concerns the consistency of the study.

- **AR**: Thank you for raising this important point, which is undoubtedly of great importance and must be clarified accordingly. Indeed, the AC have maintained the same structure and composition throughout the full studied period. We have shown this by adding an additional panel in Figure 2 (See Panel b Figure 2) with up to 6 examples of AC distributed throughout the full period. In these examples, we can observe how the structure and composition of the ACs have barely changed through time. This is so because the ACs were official documentation that had to be endorsed by official state paper. As such, the government required (and it was mandatory by law) that the structure should be the same, and that it should be consistent through time. In this regard, we have added explanatory text in the manuscript (See page 6 lines 13 to 24).

However, the amount of paper used each year exhibits an important inter-annual variability. This variability could be related to the variable amount of issues that needed to be addressed by local authorities in a given period. In addition, it could also be dependent of the available budget of the municipality, since more money available implies a less restrictive use of sealed paper. Still, this year-to-year variability is not likely to cause systematic biases in the methodology, as it does not present any temporal behavior that could affect the trends detected in the variables Drought and extreme rainfall (see Figure A1 in Appendix A ). The fact that the variability in the use of sealed paper does not affect drought and extreme rainfall estimation using the COST method is further demonstrated when observing that there is no significant statistical correlation between the amount of sealed paper and the percentage used to inform about droughts and extreme rains (See Figure A2 in Appendix A). Even more clearly, in anomalous years with the largest (fewest) use of sealed paper, there is no anomaly whatsoever in the amount of paper dedicated to inform about droughts or extreme rainfalls. Regarding a possible seasonality of the use of sealed paper, the law dictated that the meetings of the Cabildo should be held once a week, regardless of the time of the year. For this reason, the use of sealed paper should not present any obvious seasonality, having every month a similar amount of used paper. November and December show a somewhat lower use of sealed paper (See Figure A3 in Appendix A ), but even in these months, and from the examples used in Figure 2 (Panel B), we observe that the amount of sealed paper used to discuss climatic events is never below 4% of the total. Therefore, we can conclude that the percentage of the sealed paper used exhibits no strong seasonality.

Certainly, the rule of celebrating weekly meetings was not allays fulfilled, and the number of meetings was affected by the urgency of the topics to address and with administrative matters. This is, it is clear that the frequency and importance of the meetings responded to the daily problems of the municipalities. For this reason, there are years with several weeks without meetings, while sometimes several meetings were held within the same week. In any case, there is no seasonality in the amount of AC, as evidenced by similar studies in other areas of Spain (Pérez, 1987, Gutiérrez, 2005). Anyway, the most important fact is that there was no period of the year when the town hall meetings should stop. Therefore, the importance of the issues and the conjuncture of a given year explain variations in the amount of paper used in a specific year, but it is always sure that if something extraordinary happened (such as lack of water or heavy rainfall), the town hall met to discuss the details regardless of the date and epoch of the year. The fact that the COST method offers the data as a percentage of the total annual paper sheets AC, normalizes the paper difference between the different months and seasons. Therefore, we consider that the COST method is valid for conducting studies with monthly and seasonal resolution.

Section 3.3 includes now part of the previous explanations about the variability in the annual amount of sealed paper, and how it does not affect the reconstructed data, as well as the fact that the amount of sealed paper used should not affect the ability of the COST method to conduct monthly or seasonal studies (See Page 1 lines 23 to 34).

Finally, we have prepared a new Apendice (See See Apendice A) which contains 3 figures, with the goal of showing how the annual variability of the amount of sealed paper exhibits no statistically significant trend and therefore does not affect the trends detected in the reconstructed events (Figure A1 in Appendix A ). Figure A2 in Appendix A, shows that there is no statistical correlation between the annual amount of paper and the percentage of paper used to talk about droughts and extreme rainfall by the COST method. Finally, Figure A3

in Appendix A, shows how the ACs do not present seasonality, nor monthly bias that could affect the validity of the COST method to perform reconstructions at monthly or seasonal time scales.

- **RC**: Regarding the methodology, there are several important details that should be clearly stated in the text instead of being included in the tables and figures captions. The reading and comprehension of paper is difficulted by this fact, in my opinion. I suggest an improvement of section 3, providing a more clear and detailed description of all methodological procedures undertaken through the study.
- **AR**: Thanks for your comment, which overlaps to some extent with the comments by Reviewer #1. We have worked to make clearer explanations of each methodology. In this version of the manuscript, Section 3 has been modified including now several clarifications regarding the steps followed in each method (page 9 lines 10 to 18 for the Rogations method; page 10 lines 8 to 11, page 11 lines 1 to 3 and lines 23 to 27, and page 12 lines 1 and 2, and lines 14 to 16 for the Content Analysis method; page 14 lines 22 to 35 and page 13 lines 4 to 7 and lines 12 to 19 for the COST method). Additionally, Figures 3, 4 and 5, include now explanatory panels showing the flow chart for each method and the examples in figures are now more detailed along the text of Section 3.

- **RC**: There are some important problems detected in Figures as follows: a) Title of Figure 7 (drought variability) is not suitable e should be modified according the Figure 8 title (extreme rainfall variability). b) Figure 9 is not legible and must be resized.
- **AR:** These errors have been corrected by modifying the title of Figure 7 and resizing Figure 9.

**Minor comments.**

- **RC**: 1) The text needs a general revision of the English. There are several unclear expressions, some mistakes and missing words. I suggest a general revision of the text redaction. 2) The titles of sections 2 and 3 should be modified because the authors should point out the sources and methods used in their own study and not in such general mode as "Sources in Historical Climatology". In my opinion this is incorrect. 3) The final section must be a "Conclusion" instead of "Results" (repeated section title).
- **AR**: All the mistakes have been corrected. Further, and pointed out in the response to reviewer #1, the text has been carefully reviewer by a professional translator.
- Here is a more detailed description of the changes carried out:

    1) We have made a deep revision of the language in order to improve their understanding, shortening some sentences and try to improve the redaction style. In addition a native English speaker has perform a full revision of the manuscript.

    2) Section 2 has been renamed to "Documentary sources" and Section 3 to "Methodology".

    3) We have corrected the name of Section 5, now is "Conclusions", as pointed out by the reviewer.

[revised manuscript text omitted]

---

## Author Response (AR2)

**Editor Decision:** Publish subject to technical corrections (18 Jun 2019) by Chantal Camenisch
Comments to the Author:
Many thanks for your revised paper. It looks very nice.

5   Please check, if the caption of figure 9 is correct.

Yours,

10   **Authors' response:**

We are very happy to know that the paper will be accepted for publication in CP. Thank you very much
for your work and kind help.
Thanks for detecting the error in Figure 9.
15   The figure caption is correct, but in the Figure 9, a1 should be replaced by a2
and vice versa. In the next revised version of the manuscript, we have replaced Figure 9, by a Figure 9
with panels title corrected. This change is marked with control change. Following your instructions, this
has been the only necessary change made in the manuscript.

[revised manuscript text omitted]